# Structural mechanism of BRD4-NUT and p300 bipartite interaction in propagating aberrant gene transcription in chromatin in NUT carcinoma

Di Yu [1,2,5], Yingying Liang[1,2,5], Claudia Kim[3], Anbalagan Jaganathan[3], Donglei Ji [1,2], Xinye Han[1,2], Xuelan Yang[1,2], Yanjie Jia[1], Ruirui Gu[1,2], Chunyu Wang[4], Qiang Zhang[1], Ka Lung Cheung[3], Ming-Ming Zhou [3]✉ & Lei Zeng [1,2]✉

BRD4-NUT, a driver fusion mutant in rare and highly aggressive NUT carcinoma, acts in aberrant transcription of anti-differentiation genes by recruiting histone acetyltransferase (HAT) p300 and promoting p300-driven histone hyperacetylation and nuclear condensation in chromatin. However, the molecular basis of how BRD4-NUT recruits and activates p300 remains elusive. Here, we report that BRD4-NUT contains two transactivation domains (TADs) in NUT that bind to the TAZ2 domain in p300. Our NMR structures reveal that NUT TADs adopt amphipathic helices when bound to the four-helical bundle TAZ2 domain. The NUT protein forms liquid-like droplets in-vitro that are enhanced by TAZ2 binding in 1:2 stoichiometry. The TAD/TAZ2 bipartite binding in BRD4-NUT/p300 triggers allosteric activation of p300 and acetylation-driven liquid-like condensation on chromatin that comprise histone H3 lysine 27 and 18 acetylation and transcription proteins BRD4L/S, CDK9, MED1, and RNA polymerase II. The BRD4-NUT/p300 chromatin condensation is key for activating transcription of pro-proliferation genes such as *ALX1*, resulting ALX1/Snail signaling and epithelial-to-mesenchymal transition. Our study provides a previously underappreciated structural mechanism illuminating BRD4-NUT's bipartite p300 recruitment and activation in NUT carcinoma that nucleates a feed-forward loop for propagating histone hyperacetylation and chromatin condensation to sustain aberrant anti-differentiation gene transcription and perpetual tumor cell growth.

Gene fusions, resulting from chromosomal rearrangements, are strong driver mutations present in a wide array of cancers[1]. They represent first-line indicators for diagnosis, prognosis, and cancer subtype biomarkers, and serve as targets for drug development. Currently, over 25,000 fusions have been identified in 33 cancer types and 16.5%

cancer cases[2]. In hematological disorders, most recurrent gene fusions are studied, such as *BCR-ABL1* in chronic myeloid leukemia (CML)[3], *PML-RARA* in acute promyelocytic leukemia (APL)[4], *RUNX1-RUNX1T1* in acute myeloid leukemia (AML)[5] and *SIL-TAL1* in pediatric T-cell acute lymphoblastic leukemia (T-ALL)[6]. Functional consequences of gene

fusions include dysregulation of parent gene expression and chromosome aberrations by fusion genes[1,7]. However, for solid tumors (mesenchymal and epithelial), disease-related molecular signatures of fusion genes and their encoded proteins remain poorly understood[8].

Nuclear protein in testis (NUT) carcinoma (NC, or NUT-midline carcinoma, NMC) is a rare and one of the most aggressive solid tumors with a median survival of 6.5 months[9]. It often arises in the midline regions (head, neck, lungs, or thorax), and affects primarily adolescents and young adults[10]. The genetic hallmark of NC is a rearrangement of *NUT* on chromosome 15q14 that is fused with *BRD4* (but in some cases *BRD3*, *NSD3*, or other genes), creating a fusion gene that encodes BRD4-NUT fusion protein (Fig. 1a)[11,12]. Unlike many gene fusions that directly alter gene expression, BRD4-NUT acts as a transcription co-activator. It binds to lysine-acetylated histones through BRD4 tandem bromodomains (BrDs) and recruits histone acetyltransferase (HAT) p300/CBP to chromatin through the NUT moiety[13,14] (Fig. 1a). The p300/CBP recruitment induces excessive histone H3 lysine 27 acetylation (hyper-acetylation; H3K27ac) and creates unusually large topological domains (aka megadomains) in chromatin that enable aberrant gene transcription by sequestering transcriptional activators away from pro-differentiation genes[12–17] and activating proliferation and anti-differentiation genes[16–21], resulting in perpetual tumor cell growth. Notably, p300/CBP recruitment does not always correlate with chromatin acetylation including H3K27ac and gene activation[22,23]. Therefore, it is imperative to examine the mechanism that regulates BRD4-NUT recruitment of p300/CBP and the resulting activation of p300/CBP HAT activity in cellular progression of NC.

The histone acetyltransferase p300/CBP, contains both structured and disordered domains, including nuclear receptor interaction domain (NRID, disordered in free state), transcription adapter zinc binding domains (TAZ1, TAZ2), and domains such as KIX, BrD, PHD, HAT, and ZZ (Fig. 1a)[24–27]. The TAZ2 domain is an adaptor protein module that enables p300 to nucleate the assembly of diverse transcription factors (TFs) into multicomponent co-activator complexes including p53, IRF3, STAT3, and MEF2[28–31]. As shown previously[27,31,32], p300 normally is in a dimeric and autoinhibitory state for its HAT activity, and gets activated upon binding to phosphorylated and dimerized transcription factors (IRF3 and STAT1) on chromatin. The p300/transcription factor binding-induced p300 activation is shown by p300 *trans*-autoacetylation and its substrate acetylation of histone H3K18 and H3K27 in chromatin[31]. Although it has been reported that the F1c region in NUT of BRD4-NUT interacts with the TAZ2 domain of p300 (Fig. 1a)[13], the structural and molecular details of the F1c/TAZ2 interaction and its relation to p300 activation, histone hyperacetylation, and aberrant gene transcription for NUT carcinoma cell proliferation have remained elusive[13].

In this study, we report that BRD4-NUT contains two transactivation domains (TADs) in NUT that bind to the four-helical bundle TAZ2 domain of p300. The BRD4-NUT/p300 bipartite binding activates p300 HAT and results in histone H3 lysine 27 and 18 hyperactivation, leading to formation and spread of BRD4-NUT/p300 nuclear condensation on chromatin that comprise key transcription proteins BRD4L/S, CDK9, MED1, and RNA polymerase II. We further show that the BRD4-NUT/p300 chromatin condensation is important for transcriptional activation of pro-proliferation genes such as *ALX1*, a highly expressed oncogene in NUT carcinoma cells, and resulting ALX1/Snail signaling and epithelial-to-mesenchymal transition.

## Results

### BRD4-NUT F1c domain constitutes two binding motifs for p300 TAZ2 domain

The F1c region in BRD4-NUT consists of intrinsically disordered regions (IDRs)[33] and an ordered segment comprising residues 407–441 (Fig. 1a, b), which was predicted to have α-helical propensity by partition and semi-random subspace method (PSRSM)[34]. Our GST pull-

down study with different GST-F1c protein fragments (Fig. 1c) confirmed that the N-terminal fragment of F1c (residues 346–480) binds to the MBP-TAZ2 domain of p300 (Fig. 1d, lanes 2 and 3)[13]. Surprisingly, two separate fragments (residues 377–418 and 419–470) can bind to the TAZ2 domain (Fig. 1d, lanes 5 and 6). Our $^1$H-$^{15}$N heteronuclear single-quantum coherence (HSQC) NMR study confirmed that the p300 TAZ2 domain displayed spectral perturbations upon addition of NUT fragments of residues 377–418 or 419–470 (Fig. 1e) but not for residues 346–376 and 470–592 (Supplementary Fig. 1a). Notably, the 419–470 sequence induced similar but more dispersed resonance changes of the TAZ2 domain than the 377–418 sequence (Supplementary Fig. 1b), suggesting that the former engages in more extensive interactions. Nevertheless, the two sequences have similar affinities to TAZ2 ($K_d$-32 μM for residues 377–418, and $K_d$-36 μM for residues 419–470) (Fig. 1f), indicating their equal importance for complex formation with p300 TAZ2. Finally, size-exclusion chromatography (SEC) analysis showed that free His-F1c (residues 346–592) existed as an oligomer (MW-155 kDa), and formed a high molecular weight complex (MW-246 kDa) with MBP-TAZ2 (MW-63 kDa) (Fig. 1g, Supplementary Fig. 1c), indicating that the BRD4-NUT/p300 protein complex likely exists as oligomers.

We then used synthetic peptides to define the F1c binding motifs (Supplementary Table 1). Addition of the F1c peptide (residues 403–418) caused characteristic but less chemical shift perturbations in the HSQC spectrum of the $^{15}$N-TAZ2 domain than the longer 377–418 sequence (Fig. 1h left panel vs. Fig. 1e left panel, Supplementary Fig. 2a), which is consistent with their binding affinity, i.e., $K_d$-115 μM for residues 403–418 vs. $K_d$-32 μM for residues 377–418 (Fig. 1f, Supplementary Fig. 2a). These results indicate that the 403–418 peptide resembles the essential binding motif for the 377–418 sequence. Another F1c peptide (residues 435–454) did not exhibit much binding to the TAZ2 domain as shown by HSQC spectra and weak affinity ($K_d$-958 μM) (Supplementary Fig. 2b). Accordingly, we engineered a fusion TAZ2-NUT protein by joining the TAZ2 domain (residues 1723–1812) to NUT (residues 419–470) with a flexible six-residue Gly-Ser repeat sequence (Fig. 1c). The NMR chemical shift differences between $^{15}$N-TAZ2-NUT and $^{15}$N-TAZ2 (free state) are nearly identical to those of $^{15}$N-TAZ2 upon addition of NUT (residues 419–470) (Fig. 1h right panel, Supplementary Fig. 2c). Therefore, our engineered TAZ2-NUT protein represents an accurate binding model of the TAZ2 and NUT (residues 419–470) complex, allowing us to determine the high-resolution structure of this complex.

### Structures of the p300 TAZ2 domain/BRD4-NUT F1c complex

We determined the three-dimensional (3D) structures of the p300 TAZ2 domain (residues 1723–1812) in complex with a F1c peptide (residues 403–418) and the TAZ2-NUT protein (TAZ2, residues 1723–1812; NUT, residues 419–470) using triple-resonance NMR spectroscopy methods (Supplementary Fig. 3a, b; Supplementary Table 2)[35]. The NUT F1c residues 403–418 and residues 419–470 are referred as the transactivation domains 1 and 2 (TAD1 and TAD2) of NUT, respectively, in the rest of this study. Final RMSDs of 0.40 +/− 0.095 Å and 0.33 +/− 0.084 Å for backbone atoms of the TAZ2 protein complexes, respectively, indicate that these structures are well defined (Supplementary Table 2). The TAZ2 structure consists of a four-α-helix bundle with three zinc atoms bound in the interhelical loops (Fig. 2a, b). The complex of TAZ2/TAD1 shows that the NUT TAD1 peptide retains α-helical conformation and their side chains interact with the TAZ2 surface, which are similar to the soluble structure of p300 TAZ2 in complex with the p53 TAD2 peptide (Fig. 2a vs. 2c; PDB: 2MZD)[36]. The F1c residues Y405, L410, L411, Y413, and I414 bind to the hydrophobic patch of TAZ2, formed by the side chains of I1735, A1738, K1760, M1761, R1763, V1764, I1781, Q1784, L1785, and L1788, at the interface between α1, α2, and α3 helices of TAZ2 (Fig. 2a). Specifically, L410 side chain is anchored into the hydrophobic cavity of TAZ2, Y413

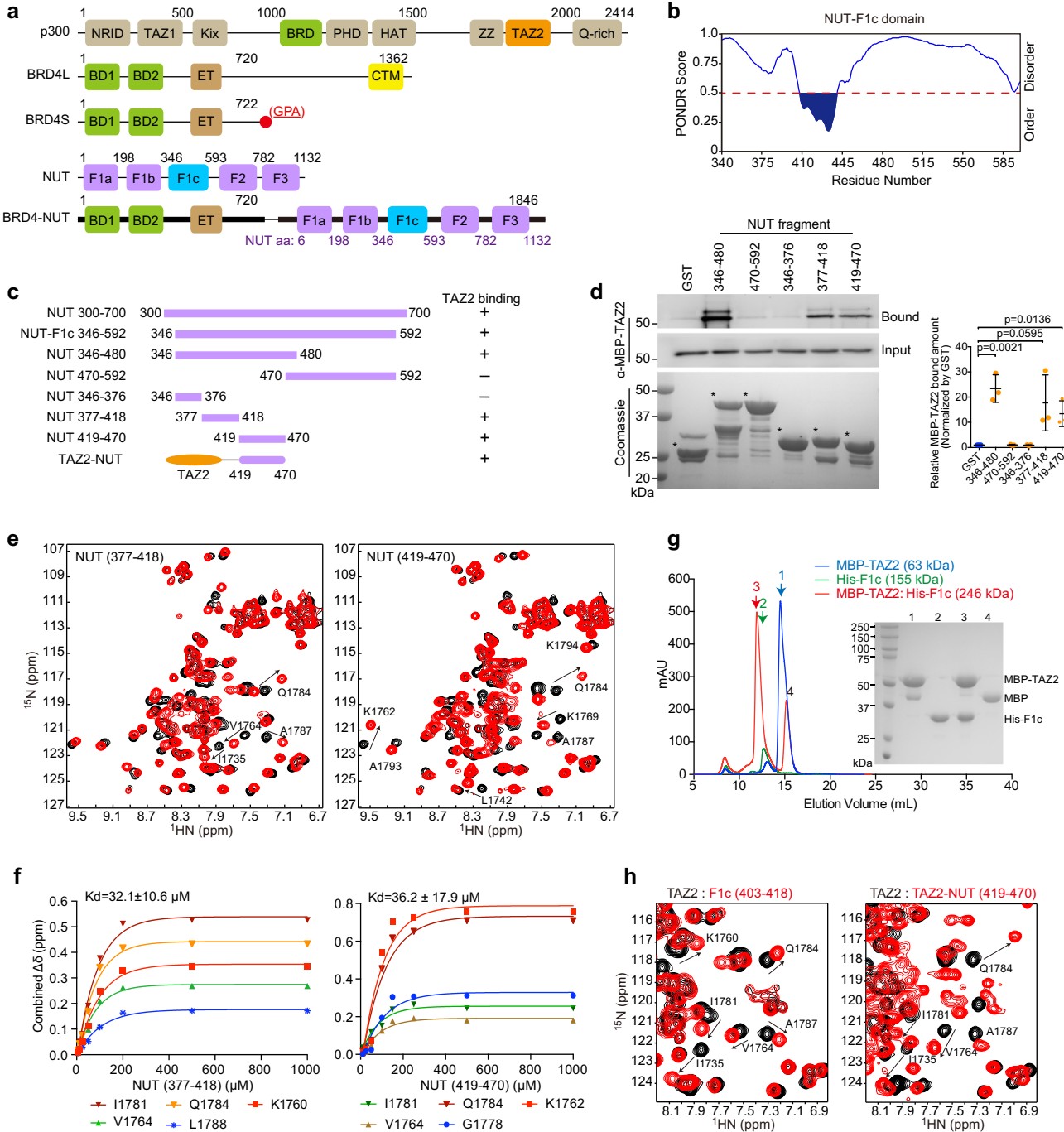

**Fig. 1 | NUT F1c region contains two TADs that bind to p300 TAZ2 domain.**
**a** Schematic diagram showing protein domains of p300, BRD4 isoforms, NUT, and BRD4-NUT fusion protein. **b** Plot showing order/disorder prediction of the BRD4-NUT F1C region by the PONDR algorithm. **c** Various recombinant NUT protein constructs used in the binding assay. **d** GST or GST-NUT fragments were used to pull down MBP tagged p300 TAZ2 domain. Both NUT TAD1 (residues 377–418) and TAD2 (residues 419–470) sequences can equally bind to TAZ2 domain. The proteins were visualized using antibodies as indicated. Data represent mean +/− SEM ($n$ = minimum of three independent experiments for each condition). $p$-values for each indicated comparison are two-tailed unpaired and derived from Student's $t$-test without adjustment for multiple comparisons. **e** Overlays of NMR $^1$H-$^{15}$N-HSQC spectra of the $^{15}$N-labeled TAZ2 domain in the free form (black) or in the presence of the NUT fragments as indicated (red). The TAZ2/NUT molar ratio is 1:3 (red). Peaks that exhibit significant chemical shift perturbations are marked with residue numbers. Arrows indicate chemical shift changes for individual peaks. **f** The HSQC

binding analysis of TAZ2 domain with NUT fragments as indicated. The saturation curves were calculated using a single-site binding equilibrium model. **g** Size-exclusion chromatography reveals oligomeric forms of His-F1c (residues 346–592) and MBP-TAZ2/His-F1c mixture. The right panel shows SDS-PAGE analysis and silver staining of individual fractions. Reference proteins were also subjected to the same chromatographic treatment for comparison. The elution profiles are Red: MBP-TAZ2/His-F1c protein complex in the 1:1 mix ratio; blue: MBP-TAZ2; green: His-F1c with apparent molecular mass of 246, 63 and 155 kDa, respectively. The calculated molecular masses by SDS-PAGE of His-F1c, MBP and MBP-TAZ2 monomers are 30, 42 and 60 kDa, respectively. His-F1c assumes a 155-kDa oligomer comprised of 5 subunits of monomer (MW~30 kDa). Experiments were repeated at least twice. **h** Overlay of the HSQC spectra of the TAZ2 domain in the free form (black) and in the presence of the F1c peptide (red) (residues 403–418) (left) or TAZ2-NUT fusion protein (right).

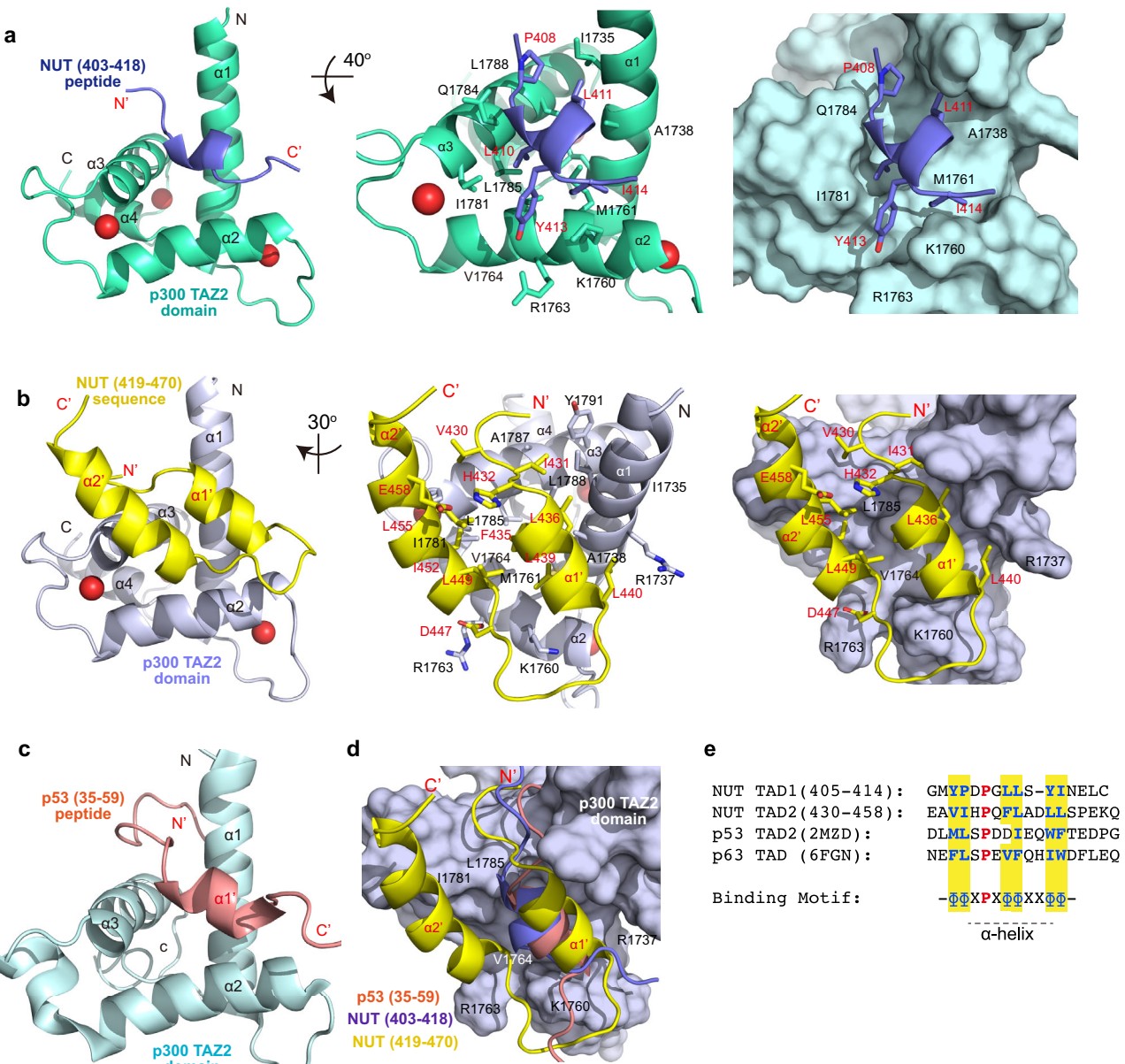

**Fig. 2 | Structures of p300 TAZ2 domain in complex with NUT TAD1 or TAD2.**
**a** Interaction between p300 TAZ2 domain and F1c peptide (TAD1, residues 403–418). Left: Ribbon depiction of the lowest energy structure of the complex; Middle: Expanded diagram of key inter-molecular interactions from the α-helix F1c peptide to the hydrophobic binding core of the TAZ2 domain surrounded by α1, α2, and α3 helices. TAZ2 domain residues involved in peptide binding are labeled and colored in light-green, the peptide elements are colored in blue; Right: Surface representation of the TAZ2 domain depicting its recognition to the F1c peptide in the same orientation as (Middle). **b** Structure of TAZ2-NUT fusion protein containing the NUT sequence (TAD2, residues 419–470). Left: Ribbon depiction of the lowest energy structure of the fusion protein (TAZ2 in light-blue and NUT sequence in yellow); Middle: The binding core depiction of residue side-chains of TAZ2 in α1, α2, and α3 helices for hydrophobic and electrostatic interactions to the residues in the F1c sequence. Right: Surface representation of interaction positions of TAZ2-NUT fusion in the same orientation as (Middle); **c** The ribbon diagram of CBP TAZ2/ p53 TAD2 complex. **d** Superposition of NUT TAD2 sequence (residues 419–470, in yellow) in the TAZ2-NUT fusion protein with NUT TAD1 sequence (residues 403–418, in blue), and p53 TAD2 (residues 35–59, in orange, 2MZD) peptides. **e** Sequence alignment of NUT TAD1 (residues 405–414), NUT TAD2 (residues 430–458), p53 TAD2, and p63 TAD. The conserved proline amino acid is colored in red, hydrophobic residues in blue. The binding motif common to these sequences is diagramed as, Φ represents a hydrophobic amino acid and X represents any amino acid.

aromatic ring of F1c packs against side chains of positively charged K1760 and R1763 of helix α2, whereas Y405, P408, and L411 side chains participate in an extensive network of hydrophobic interactions with TAZ2 residues on α1 and α3 helices. In addition, mutation of L410G, L411G, Y413G. and I414G abrogated the peptide binding (Supplementary Fig. 3c), validating the NUT TAD1 motif (residues 403–418) for the TAZ2 interaction.

In the TAZ2-NUT TAD2 structure, the fused NUT sequence of residues 433–460 forms two anti-parallel α-helices. The first α-helix

(α1', residues 433–440) is in a similar position as the TAD1 peptide (residues 403–418), whereas the second α-helix (α2', residues 448–460) flips back to interact with the first α-helix (α1', residues 433–440) and the α2-helix of TAZ2 protein surface (Fig. 2b vs. 2a). For example, side chains of F435, L436, L439, L440 from the α1'-helix are situated in the TAZ2 binding pocket. P448, L449, L451, I452, and L455 from the α2'-helix interact with TAZ2 residues on the α2 and α3 helices and the TAD1 α1'-helix (Fig. 2b). Side chains of preceding residues A429, V430, and I431 participate in hydrophobic interactions with

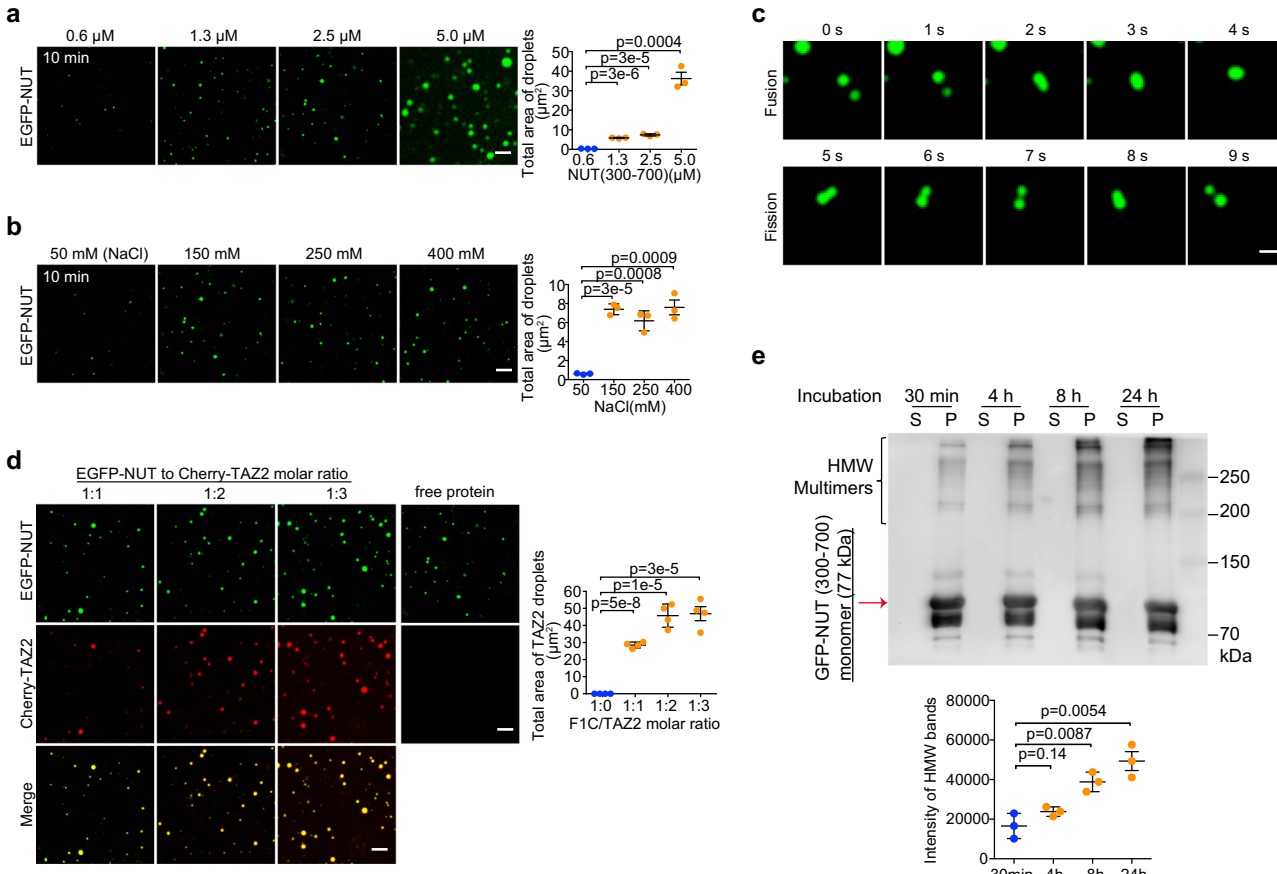

**Fig. 3 | NUT droplet formation is facilitated by its binding to p300 TAZ2.**
**a** Fluorescence images showing liquid droplets of EGFP-NUT (residues 300–700) with increasing amounts of the protein as indicated. Histogram plots showing total area of droplets per field (36 × 36 μm). Scalar bars: 5 μm. Quantitative data (right) are shown as mean +/− S.D. (*n* = minimum of three independent samples for each condition). *p*-values for each indicated comparison are two-tailed unpaired and derived from Student's *t*-test without adjustment for multiple comparisons.
**b** Microscopic images depicting EGFP-NUT (residues 300–700, 2.5 μM) droplet formation at different salt concentrations as indicated. Scalar bars: 5 μm. Quantitative data (right) are shown as mean +/− S.D. (*n* = minimum of three independent samples for each condition). *p*-values for each indicated comparison are two-tailed unpaired and derived from Student's *t*-test without adjustment for multiple comparisons. **c** Images showing fusion (top) or fission (bottom) of EGFP-NUT (residues 300–700, 2.5 μM) droplets in a time course as indicated. Scalar bars: 1 μm. Experiments were repeated at least twice. **d** Fluorescence images showing liquid

droplets of EGFP-NUT (residues 300–700, 2.5 μM) with addition of Cherry-TAZ2 as indicated. Histogram plots showing total droplet areas per field (36 × 36 μm) in different NUT to TAZ2 molar ratio. Three fields are quantified. Scalar bars: 5 μm. Quantitative data are shown as mean +/− S.D. (*n* = minimum of four independent samples for each condition). *p*-values for each indicated comparison are two-tailed unpaired and derived from Student's *t*-test without adjustment for multiple comparisons. **e** Supernatant (S) and pelleted (P) fractions from droplet spin down experiments were run in western blots to assess the time-dependent (0–24 h of phase separation) formation of heat-, reducing- and SDS-stable EGFP-NUT/Cherry-TAZ2 oligomers. Note that some high molecular weight protein complexes (HMW multimers) appeared shortly after addition of Cherry-TAZ2. Data represent mean +/− SEM from three independent experiments. *p*-values for each indicated comparison are two-tailed unpaired and derived from Student's *t*-test without adjustment for multiple comparisons.

residues located on α1 and α3 helices of TAZ2. Together, they establish more extensive contact with the TAZ2 surface (Fig. 2b). Side chain of D447 of NUT also forms electrostatic interactions with R1763 of TAZ2, and the carbonyl group of D438 backbone is within hydrogen-bonding distance with the side chain of K1760 of TAZ2 domain (Fig. 2b). Both NUT TAD1 and TAD2 sequences adopt a similar conformation on the TAZ2 domain as compared to the p53-TAD2 structure (Fig. 2c, d), indicating that TAZ2 functions as an adaptor to recruit transcription proteins. Alignment of NUT TAD1 and TAD2 residues 403–418 and 428–445 with p53-TAD2 or p63-TAD peptides reveals consensus binding sequence of ΦΦxPxΦΦxxΦΦ (where P represents proline, Φ is a hydrophobic amino acid, and x any amino acid; Fig. 2e)[36].

### NUT liquid-like condensation is enhanced by the p300 TAZ2 incorporation
Recent studies from us and others showed that protein IDRs mediate liquid-like condensation in the nuclear assembly for gene transcription[37–39]. We investigated the condensation capability of

NUT IDRs in the constructs with an EGFP tag (A206K) (outlined in Fig. 1c). Purified recombinant EGFP-NUT (residues 300–700) protein formed spherical droplets in a concentration-dependent manner (Fig. 3a). Fluorescence microscopy of the opaque EGFP-NUT solutions revealed that the GFP-positive droplets appeared stable as salt concentration increased to examine the contribution of electrostatic interactions (Fig. 3b). The micron-sized droplets fused, coalesced and dripped (Fig. 3c), but dispersed following the treatment of 1,6-hexanediol that is known to perturb weakly hydrophobic interactions (Supplementary Fig. 4a). We next prepared a shorter EGFP-F1c segment (residues 346–592), which consisted of key α-helical binding motifs but with truncated IDRs as compared to EGFP-NUT protein (residues 300–700) (Fig. 1c). Microscopic images showed that this EGFP-F1c protein produced smaller and less abundant liquid droplets, which had the higher threshold concentration (10 μM) and were more sensitive to salt than EGFP-NUT droplets (0.6 μM) (Supplementary Fig. 4b, c vs. Fig. 3a), highlighting the importance of IDRs

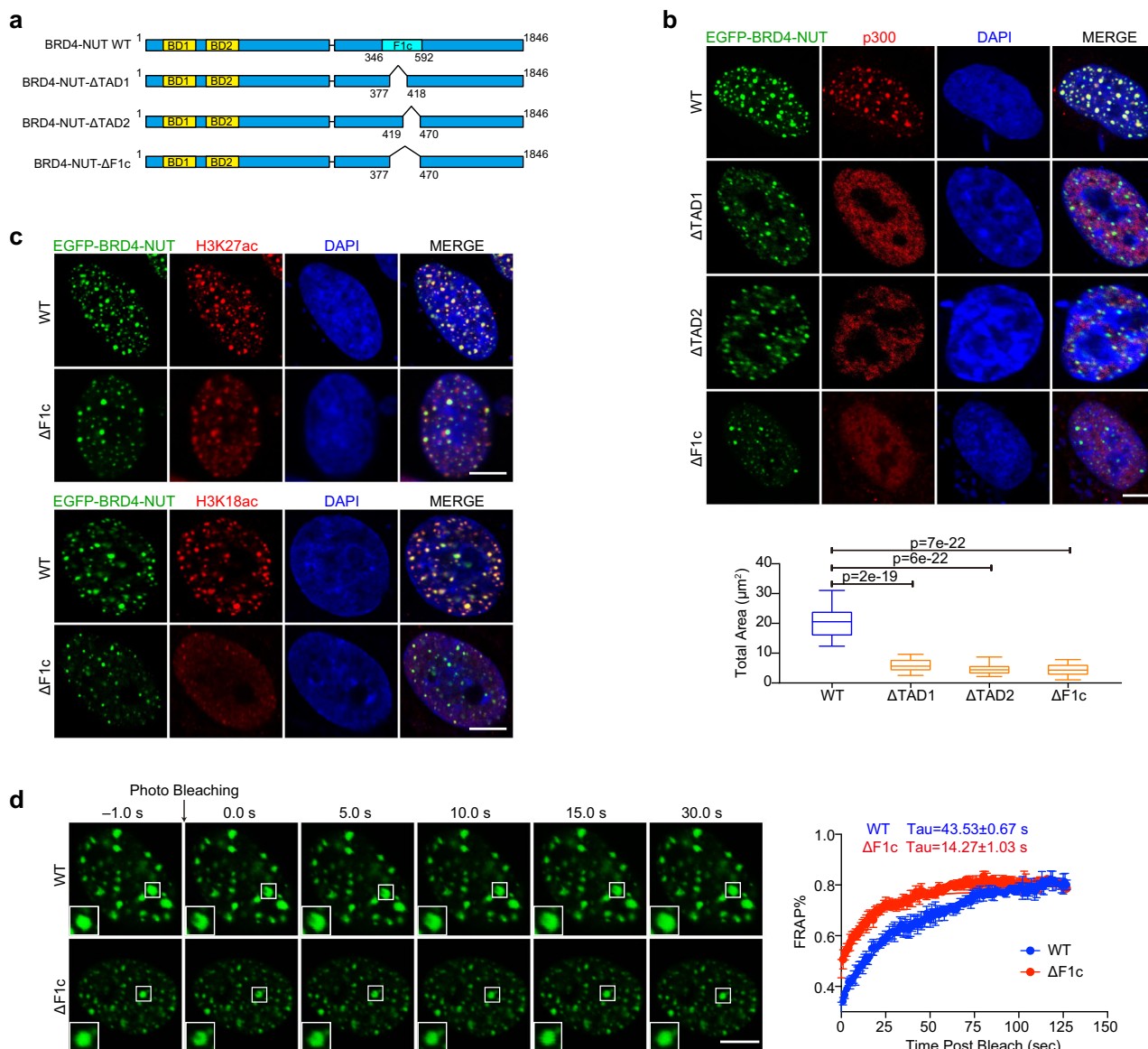

**Fig. 4 | BRD4-NUT puncta are co-localized with p300. a** Schematic representations showing domain architecture of BRD4-NUT-WT, -ΔF1c, -ΔTAD1, or -ΔTAD2 deletion mutants. **b** Microscopic images showing EGFP-BRD4-NUT-WT, -ΔF1c, -ΔTAD1, or -ΔTAD2 mutant puncta in colocalization with p300 in LO2 cell nuclei. Endogenous p300 was visualized using the p300 antibody. Box plots represent the lowest, lower quartile, median, upper quartile, and highest observations of puncta areas in cells expressed either with EGFP-BRD4-NUT-WT or indicated deletion mutants (*n* = minimum of 30 cells for each condition). Scalar bars: 5 μm. Two tailed unpaired Student's *t*-test without adjustment for multiple comparisons. **c** EGFP-BRD4-NUT-WT or EGFP-BRD4-NUT-ΔF1c in LO2 cell nuclei colocalizes with H3K27ac or H3K18ac. Endogenous acetylation marks were visualized using H3K27ac or H3K18ac antibody. Scalar bars: 5 μm (*n* = minimum of nine cells for each condition). **d** FRAP analysis of EGFP-BRD4-NUT-WT or EGFP-BRD4-NUT-ΔF1c puncta in LO2 cells. Representative images of FRAP experiment are shown as indicated time, Quantitative FRAP data are shown as mean +/− S.D. (*n* = minimum of three independent samples for each condition). Scalar bars: 5 μm.

in the NUT protein for promoting droplet formation. Both EGFP-NUT and EGFP-F1c droplets can form in the absence of EGFP (Supplementary Fig. 4d).

We next investigated how p300 TAZ2-binding to NUT may facilitate liquid-like condensation. Addition of Cherry-TAZ2 could incorporate and thus concentrate the droplets of EGFP-NUT (Fig. 3d vs. Fig. 3a) or EGFP-F1c (Supplementary Fig. 4b, e), while equivalent solutions with Cherry-TAZ2 alone remained homogenous (Fig. 3d). Notably, the droplet sizes and numbers were significantly enhanced after the NUT/TAZ2 molar ratio reached above 1:2 stoichiometry (Fig. 3d), suggesting that NUT bipartite motifs act as binding scaffolds to incorporate TAZ2 domain of p300 into the complex condensation[37]. We isolated droplets from the solution phase by sedimentation and determined the levels of NUT and TAZ2 in the pellet fraction by immunoblotting. Co-addition of EGFP-NUT and Cherry-TAZ2 resulted in high molecular weight (HMW) oligomerization in a time-dependent manner, indicating that their incorporation leads to enrichment of stable oligomers (Fig. 3e). Notably, the EGFP-NUT/Cherry-TAZ2 droplets were still sensitive to 1,6-hexanediol treatment (Supplementary Fig. 4f).

**BRD4-NUT forms nuclear condensates with p300 in cells**
Previous reports showed that BRD4-NUT incorporates p300 to produce hyperacetylated chromatin over massive genomic domains (aka megadomains) in NC cells[14]. The number and magnitude of these megadomains correlate with discrete nuclear foci observed in NC cells[13,14,18]. To investigate whether the BRD4-NUT fusion protein facilitates biomolecular condensation in cells, we transfected wild-type

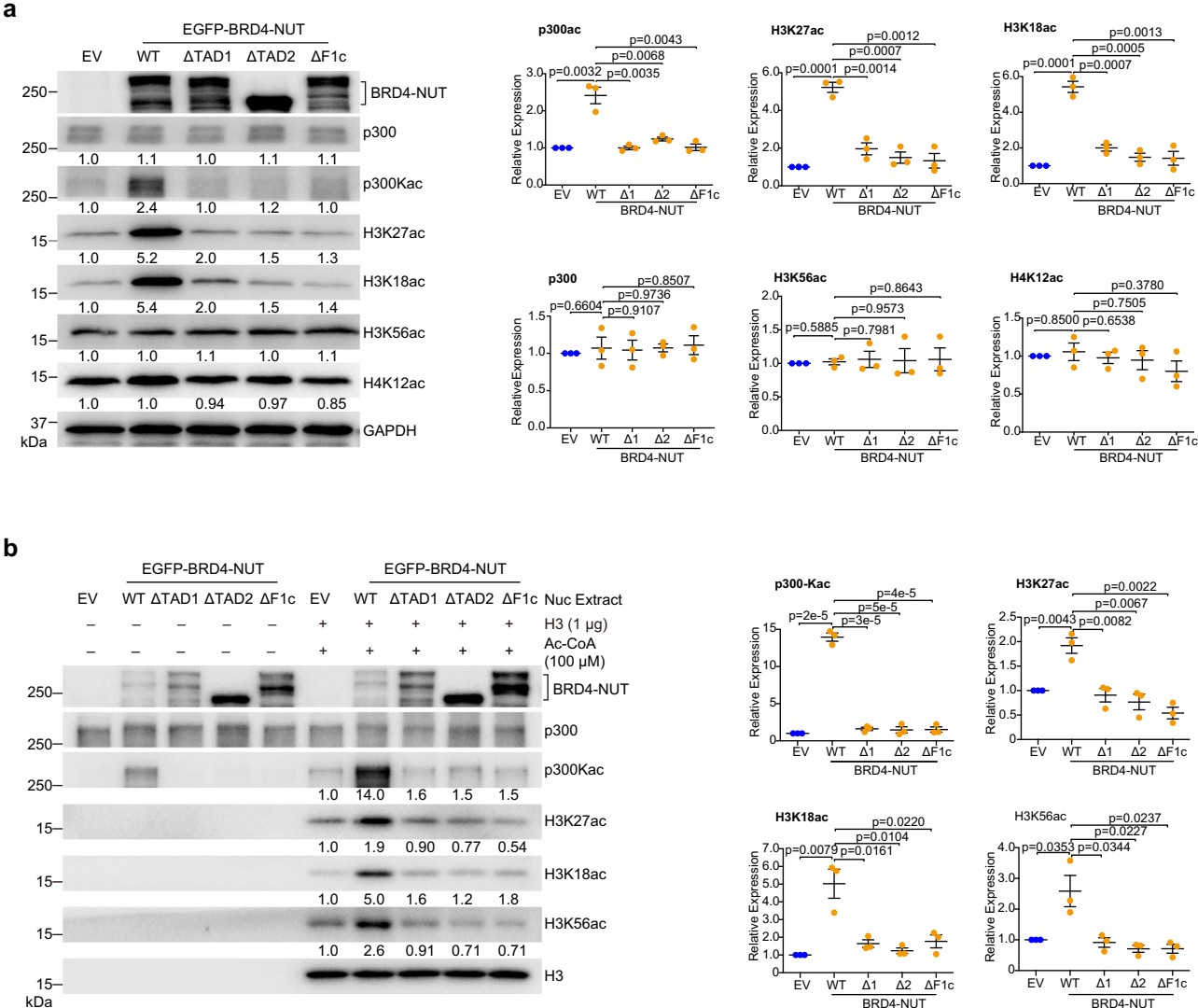

**Fig. 5 | BRD4-NUT/p300 binding activates p300 HAT activity. a** Western blot analysis showing levels of p300, acetylation of p300, acetylation of histone H3 at K18, K27 and K56, and histone H4 at K12 in the lysates of HEK293T cells transfected with EV (Empty Vector), BRD4-NUT-WT, -ΔF1c, -ΔTAD1, or -ΔTAD2 mutant. Quantitative data (right) are shown as mean +/− S.D. (*n* = minimum of three independent experiments for each condition). *p*-values for each indicated comparison are two-tailed unpaired and derived from Student's *t*-test without adjustment for multiple comparisons. The relative expression was normalized by GAPDH. **b** In vitro p300 HAT activity analysis of HEK293T cells transfected with EV, BRD4-NUT-WT, -ΔF1c,

-ΔTAD1, or -ΔTAD2 mutant. Equal amounts of nuclear extracts were used for the p300 HAT activity analysis by adding acetyl-CoA (100 μM) and histone H3 (1 μg) and visualized by antibodies as indicated. Quantitative data (right) are shown as mean +/− S.D. (*n* = minimum of three independent experiments for each condition). *p*-values for each indicated comparison are two-tailed unpaired and derived from Student's *t*-test without adjustment for multiple comparisons. The relative expressions of H3K18ac, H3K27ac, and H3K56ac were normalized by H3, p300-Kac by p300.

EGFP-BRD4-NUT-WT or TAD1/2 deletion mutants EGFP-BRD4-NUT-ΔTAD1, -ΔTAD2, or -ΔF1c (ΔTAD1/2) into LO2 (human normal liver cell line) cells (Fig. 4a). EGFP-BRD4-NUT-WT were accumulated into puncta associated with discrete nuclear foci of BRD4-NUT, p300, H3K18ac, and H3K27ac (Fig. 4b, c)[13,14]. These puncta exhibited features of biomolecular condensates, as increasing salt concentration or addition of 1,6-hexanediol in the cell culture medium caused a reduction in size and a number of nuclear puncta (Supplementary Fig. 5a). Furthermore, photo-bleaching recovery showed the evidence of dynamic internal rearrangement and internal-external exchange of molecules (Fig. 4d). We next transfected EGFP-BRD4-NUT-WT and Flag-tagged BRD4 short or long isoforms (Flag-BRD4S or Flag-BRD4L) into LO2 cells. Microscopic images revealed that ectopic Flag-BRD4S or Flag-BRD4L molecules were accumulated into EGFP-BRD4-NUT-WT puncta (Supplementary Fig. 6a)[15]. This co-localization of EGFP-BRD4-NUT-WT and Flag-BRD4S (or Flag-BRD4L) likely occurred through the

incorporation of the BRD4 moiety in BRD4-NUT with BRD4 isoforms[39]. Notably, in comparison to EGFP-BRD4-NUT-WT, ΔTAD1, ΔTAD2 or ΔF1c (ΔTAD1/2) deletion mutants displayed much less puncta formation (Fig. 4b, c) and exhibited quicker fluorescence recovery to the immobile fraction after partial photobleaching than the wild-type EGFP-BRD4-NUT-WT (ΔF1c: Tau ~14.3 s vs. WT: Tau ~43.5 s) (Fig. 4d). Furthermore, nuclear puncta of p300, H3K18ac, H3K27ac, BRD4S, and BRD4L were reduced in size and number with ΔTAD1 or/and ΔTAD2 deletion mutants (Fig. 4b, c; Supplementary Fig. 6a, b). These results indicate that depleting the bipartite interaction between BRD4-NUT and p300 decreases EGFP-BRD4-NUT nuclear foci and its co-condensation with p300, H3K18ac, H3K27ac or BRD4S/L, and increases the protein dynamics.

BRD4-NUT puncta were effectively dispersed in LO2 cells following treatment with bivalent BET-BrD inhibitor MS645 or monovalent inhibitor JQ1 (Supplementary Fig. 5a, b), confirming that inhibition of

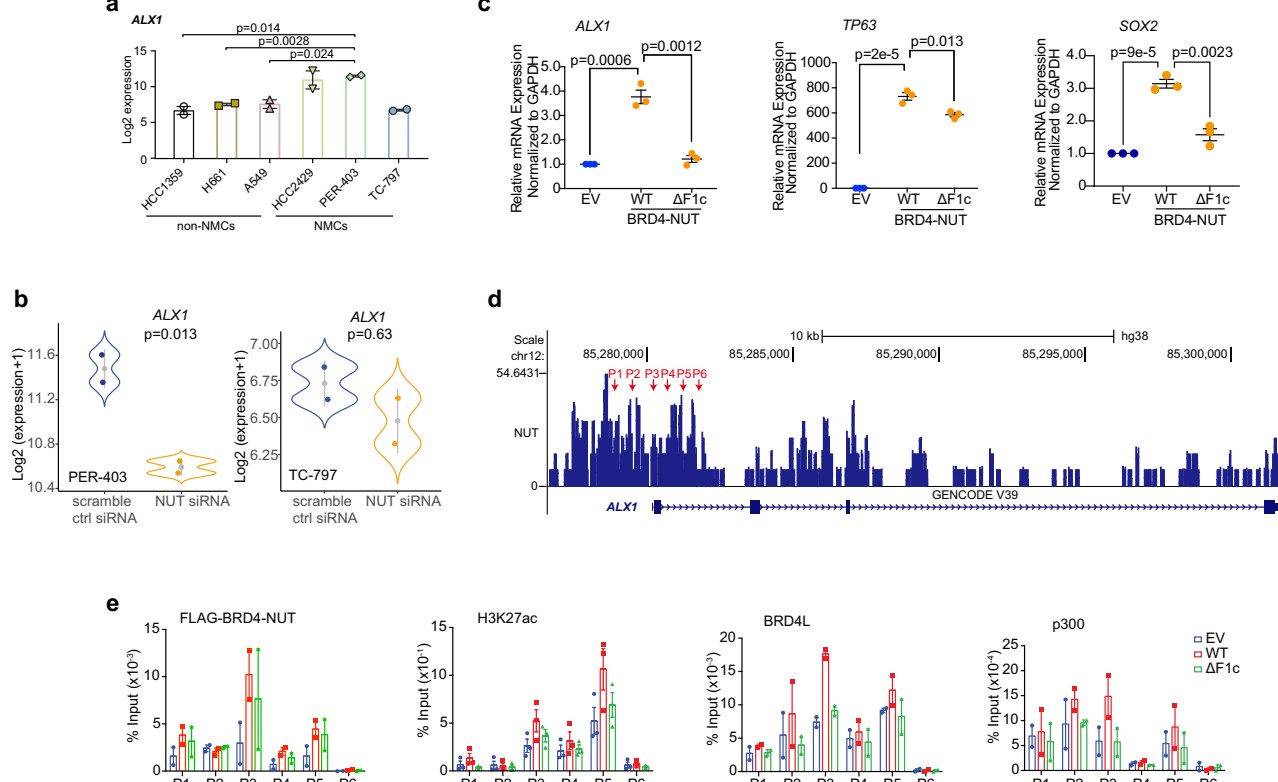

**Fig. 6 | BRD4-NUT upregulates *ALX1* transcription and promotes EMT. a** *ALX1* mRNA was significantly overexpressed in NC cells as compared to matched non-NC cells with available RNA-seq datasets (GSE14925, GSE109821, GSE199299, GSE18668) of HCC2429 and PER-403 (NC, lung), TC-797 (NC, thymus), and HCC1359, H661, and A549 (non-NC, lung). Data represent mean +/– SEM from two independent datasets for each cell-line. *p*-values are derived from two-tailed unpaired Student *t*-testing analysis without adjustment for multiple comparisons. **b** RNA-seq analysis of *ALX1* gene expressions in PER-403 and TC-797 (GSE18668) cells transfected with siRNA as indicated. Data represent mean +/– SEM from two independent datasets. The blue and orange color dots represent dataset values and gray dots indicate median values. Two-sided *p*-values for each indicated comparison are derived from the R software loaded with Limma package. **c** Relative mRNA

levels of selected genes were measured by qPCR and normalized to the GAPDH mRNA level following transfection of EGFP-BRD4-NUT-WT or EGFP-BRD4-NUT-ΔF1c in HEK293T cells. Values represent the average of three independent experiments with error bars indicating as mean +/– SEM (*n* = minimum of three independent experiments for each condition). Two-tailed unpaired Student's *t*-test without adjustment for multiple comparisons. **d** BRD4-NUT enrichment at the promoter site of *ALX1* in PER-403 cells, as shown by ChIP-seq dataset from the Gene Expression Omnibus database (GSE70870). Red arrows indicate the promoter regions of *ALX1* gene selected for the ChIP-qPCR assay. **e** ChIP analysis for enrichment of Flag-BRD4-NUT (-WT or -ΔF1c), BRD4L, H3K27ac and p300 at the *ALX1* promoter regions of HEK293T cells, as indicated in Fig. 5d. Data represent mean +/– SEM from minimum two independent experiments.

tandem BrDs binding to lysine-acetylated histones in chromatin reduced BRD4-NUT discrete condensation[40]. Notably, consistent with the previous report that histone deacetylase inhibitors (HDACis) suppress NC tumor growth in mice[41], we found that HDACis (SAHA, TMP269 or LMK235) dispersed BRD4-NUT nuclear puncta in LO2 cells overexpressing EGFP-BRD4-NUT-WT shortly after the drug treatment (12 h), but the BRD4-NUT puncta formation recurred over time (48 h) (Supplementary Fig. 5c). These results indicate that HDACis only transiently affect BRD4-NUT/p300 condensation distribution whereas BRD4 BrD inhibition by MS645 or JQ1 exerts persistent suppression of BRD4-NUT/p300 puncta formation in cells.

We further observed that transcription machinery components Mediator subunit MED1 and CDK9 formed puncta that partially overlapped with EGFP-BRD4-NUT-WT puncta (Supplementary Fig. 6a), indicating that the transcription components are drawn within proximity but may not totally colocalize with BRD4-NUT puncta. Similarly, RNA polymerase II (Pol II) phosphorylated at serine 2 (S2P) or serine 5 (S5P) was shown in nuclear puncta, which were in partial colocalization with BRD4-NUT puncta. These results agree with the previously reported limited association between BRD4-NUT and transcription components[13,15]. Notably, transfection of the deletion mutant BRD4-NUT-ΔF1c largely abolished condensation and colocalization with MED1, CDK9, RNA Pol II S2P and S5P (Supplementary Fig. 6a, b). These

results indicate that, in addition to BRD4 BrD acetyl-lysine binding, BRD4-NUT/p300 bipartite association and p300 activation are required to nucleate BRD4-NUT co-condensation with transcription proteins, and propagate histone hyperacetylation, which negatively[13,15,16] or positively[16,18–20] alters gene transcription in chromatin. The functional detail between BRD4-NUT and RNA Pol II in NC cells warrants further investigation[21].

## BRD4-NUT/p300 bipartite interaction activates its HAT activity and histone acetylation

We observed that BRD4-NUT-WT ectopically expressed in HEK293T-cells activates p300 HAT activity as reflected by markedly increased levels of p300 acetylation (p300-Kac), and p300 histone H3 substrates H3K18ac and H3K27ac[13], whereas the single TAD1 or TAD2 deletion or double deletion (ΔF1c) mutant showed nearly complete loss in their ability to activate p300 (Fig. 5a, Supplementary Fig. 7a). Further, in an in-vitro HAT assay with nuclear extracts of HEK293T cells overexpressing EGFP-BRD4-NUT-WT or the TAD deletion mutants, we confirmed that BRD4-NUT-WT increased acetylation levels of endogenous p300 (p300-Kac) and histone H3 (H3K27ac, H3K18ac, and H3K56ac), whereas neither of the single nor double TAD1/2 deletion mutants yielded any noticeable enhancement of p300 or histone H3 acetylation as compared to the empty vector control (Fig. 5b).

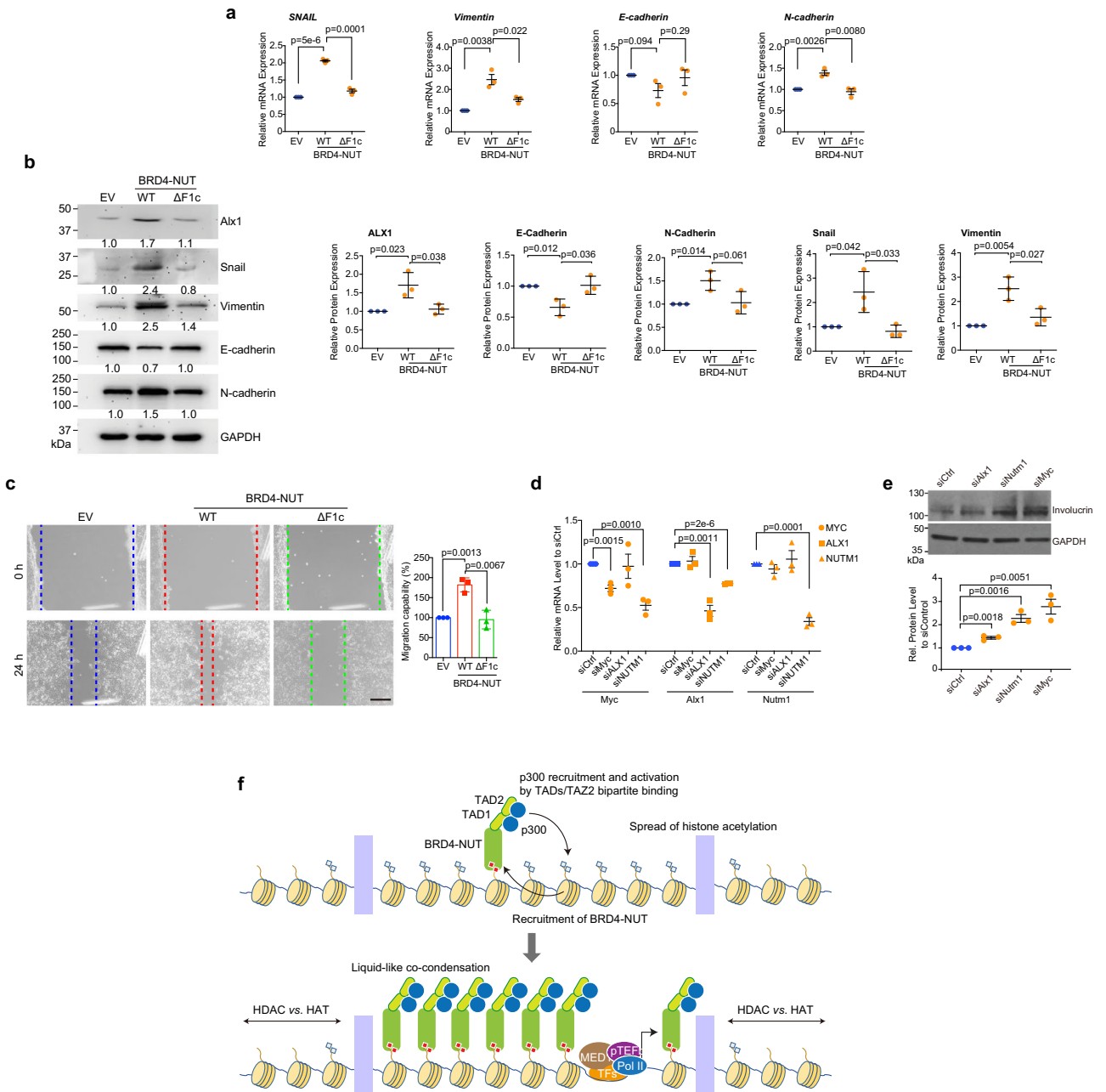

**Fig. 7 | Depletion of the BRD4-NUT/p300 interaction results in EMT signaling suppression. a** Relative mRNA expression of *Snail* and EMT markers following transfection of BRD4-NUT-WT or BRD4-NUT-ΔF1c. Bars represent mean +/− SEM (*n* = minimum of three independent experiments for each condition). Two-tailed unpaired Student's *t*-test without adjustment for multiple comparisons. **b** Western blot analysis of the expression levels of ALX1, Snail, the epithelial cell marker E-cadherin, and the mesenchymal cell markers vimentin and N-cadherin in HEK293T cell overexpressing BRD4-NUT-WT or BRD4-NUT-ΔF1c. Bars represent mean +/− SEM (*n* = minimum of three independent experiments for each condition). Two-tailed unpaired Student's *t*-test without adjustment for multiple comparisons. **c** Wound healing assay showing cell migration ability of HEk293T cells transfected with empty vector, BRD4-NUT-WT or BRD4-NUT-ΔF1c for 24 h. Migration capabilities were recorded (left) and quantitatively analyzed (right). Scalar bar: 100 μm. The data represent mean +/− SEM (*n* = minimum of three independent experiments for each condition). Two-tailed unpaired Student's *t*-test without

adjustment for multiple comparisons. **d** qPCR analysis of mRNA transcript levels of *MYC, ALX1, NUTM1* in NC 14169 cells transfected with specific siRNA targeting *Myc, Alx1, Nutm1* or control (Dharmacon) for 24 h. Bars represent mean +/− SEM (*n* = minimum of three independent experiments for each condition). Two-tailed unpaired Student's *t*-test without adjustment for multiple comparisons. **e** Western blotting analysis of protein expression levels of Involucrin and GAPDH in NC 14169 cells transfected with indicated siRNA for 72 h. The data represent mean +/− SEM (*n* = minimum of three independent experiments for each condition). Two-tailed unpaired Student's *t*-test without adjustment for multiple comparisons. **f** A model diagram illustrating a feed-forward loop mechanism propagating BRD4-NUT/p300 chromatin condensation in activating transcription of anti-differentiation and EMT genes including *ALX1* in NC cells. Deletion of the TAD1 and TAD2 domains lowers p300 occupancy at *ALX1* gene locus and results in downregulation its gene transcription.

Collectively, these results demonstrate that BRD4-NUT bipartite interaction with p300 is required for p300 HAT *trans*-activation and histone H3 hyper-acetylation in chromatin.

Given that many transcription factors including p53 directly interact with p300/CBP through their TAD domains[42], we investigated whether such tandem TADs form bipartite interactions with p300 and activate the protein. We found that TAD1 and/or TAD2 deletion in p53 reduced its interaction with p300 (Supplementary Fig. 7c), and that the wild-type EGFP-p53-WT ectopically expressed in HEK293T cells, but not its TAD1/2 deletion mutants (ΔTAD1, -ΔTAD2, or -ΔTAD1/2), increased levels of p300-Kac, H3K18ac, H3K27ac, and p53-Kac in both cellular and in-vitro acetylation assays (Supplementary Fig. 7d, e). These results confirm that p53 TADs form bipartite interactions with p300, resulting in p300 and p53 activation and histone acetylation.

### BRD4-NUT/p300 bipartite interaction promotes aberrant *ALX1* gene expression

Aristaless-like homeobox1 (ALX1) is a transcription factor that controls morphogenetic behaviors and biomineral-forming activities of skeletogenic cells[21,40]. It has been reported that ALX1 is associated with increased epithelial-to-mesenchymal transition (EMT) in cancer progression[43–45]. ALX1 promotes migration and invasion by upregulating Snail expression in lung and ovarian cancer cells[43,44]. *ALX1* knockdown suppresses cell invasion and tumor formation due to attenuated ALX1/Snail axis in lung and ovarian cancer cells[43,44], or Wnt/beta-catenin signaling pathway in melanoma cells[45]. In several NCs, including TC-797 (BRD4-NUT$^+$, thymus), PER-403 (BRD4-NUT$^+$, lung), 1015 (BRD4-NUT$^+$, lung), and tumor tissue from which 1015 was derived, massive hyperacetylated domains were observed that encompass many regulatory genes including *ALX1*[14].

We found from analyzing available RNA-seq data that *ALX1* was significantly overexpressed in NC lung cells HCC2429 and PER-403 in comparison to non-NC lung cells HCC1359, H661 and A549, whereas its expression was relatively low in NC thymus cells TC-797 (Fig. 6a). Further, knockdown of BRD4-NUT substantially downregulated the *ALX1* mRNA levels in PER-403 cells but to a lesser extent in TC-797 cells (Fig. 6b) likely due to low basal *ALX1* level. These results suggest that BRD4-NUT regulates tissue-specific transcription of *ALX1* in NC types.

To examine how BRD4-NUT/p300 facilitates *ALX1* gene expression, we transfected wild-type Flag-BRD4-NUT-WT or TAD1/2 deletion mutant Flag-BRD4-NUT-ΔF1c in HEK293T cells. Transfection of Flag-BRD4-NUT-WT stimulated transcriptional expression of *ALX1*, *SOX2*, and *TP63* (Fig. 6c), which are all important in Snail expression and squamous cancer progression[43,46]. In agreement with the results of BRD4-NUT knockdown in NC cells[47], expression of the TAD1/2 deletion mutant BRD4-NUT-ΔF1c markedly suppressed activation of *ALX1* transcription (Fig. 6c), indicating the direct role of TAD1/2 of BRD4-NUT in control of *ALX1* transcription. Notably, this TAD1/2 deletion mutant had varying inhibitory effects on *SOX2* and *TP63* expression, suggesting that additional factors may be involved in BRD4-NUT modulation of transcriptional activation of these genes (Fig. 6c; Supplementary Fig. 8a, b). Our ChIP-qPCR further confirmed that transiently expressed Flag-BRD4-NUT-WT or Flag-BRD4-NUT-ΔF1c mutant was present at the *ALX1* promoter site in HEK293T cells, and that the former correlates to enrichment levels of H3K27ac, BRD4L, and p300 at the same promoter site over the empty vector control whereas the latter is not (Fig. 6e). These results indicated that the TAD1/2 depletion undermines BRD4-NUT's ability to recruit p300 to *ALX1* gene locus, resulting in reduced histone acetylation and *ALX1* transcription.

To further investigate the role of BRD4-NUT in cell invasion and migration, we examined key EMT markers HEK293T cells transfected with wild-type BRD4-NUT or TAD1/2 deletion mutant. We found that ectopic expression of wild-type BRD4-NUT resulted in upregulation of gene transcription and protein expression of mesenchymal cell markers (Snail, vimentin and N-cadherin) and downregulation of epithelial marker E-cadherin, whereas the TAD1/2 deletion mutant had little effects (Fig. 7a, b), highlighting the importance of BRD4-NUT/p300 binding in EMT and potential epithelial morphology changes. This finding was supported by a wound-healing assay in HEK293T cells, which showed that wild-type BRD4-NUT but not the TAD1/2 deletion mutant substantially promoted cell migration capability as compared to the control cells (Fig. 7c). Finally, to directly investigate the transcriptional control of *ALX1* by BRD4-NUT/p300 in NC cells, we performed siRNA knockdown of *ALX1* and *BRD4-NUT* in NC 14169 cells, and found that siNUT reduced the transcription of *ALX1*, and both siNUT and siALX1 resulted in increased expression of Involucrin, a marker for differentiation (Fig. 7d, e), which is consistent with a previous study of NC TC-797 cells[48]. Collectively, these results highlighted that BRD4-NUT/p300 bipartite interaction is essential for transcriptional activation of *ALX1* in NC cells.

## Discussion

The identifications of recurrent gene fusions and their associated disease mechanisms in hematological disorders have facilitated the development of clinical biomarkers and new drugs targeting these fusion proteins. However, our current mechanistic understanding of fusion proteins in tumorigenesis of solid tumors is still limited. The BRD4-NUT fusion protein, a strong driver mutation in the poorly differentiated and highly aggressive NUT carcinoma[9], offers an ideal model to study the fundamental structural and molecular principles of gene fusion activities in aberrant gene transcription in chromatin in solid tumors[9,49]. In this study, we discovered that the NUT moiety in BRD4-NUT contains two TADs that adopt amphipathic helices when bound to the TAZ2 domain of p300. The BRD4-NUT bipartite binding to p300 allosterically activates p300 by relieving TAZ2 autoinhibitory function on the HAT activity, resulting in *trans*-autoacetylation of p300. This p300 activation by NUT TADs/TAZ2 binding is similar to that of p53 tandem TADs interactions with p300 as shown in our study, and also reminiscent of p300 TAZ2 domain binding to dimerized transcription factors such as STAT1 and IRF3[30,31]. Notably, the importance of the autoinhibitory function of the TAZ2 domain on p300 HAT activity is underscored by the recent reports that TAZ2 deletion mutants of p300/CBP found in many human cancers are associated with histone hyperacetylation at H3K27 and H3K18 in chromatin, two known substrates for p300/CBP[9], and recurrence following radiation in squamous cell carcinoma cohorts[50].

BRD4-NUT/p300 forms discrete nuclear condensates in chromatin through multivalent molecular interactions involving the intrinsically disordered regions in BRD4, NUT, and p300, as well as BRD4 bromodomains' binding to acetylated histones and DNA in chromatin[37–39]. Within these condensates, BRD4-NUT/p300 are colocalized with H3K27ac and BRD4L/S, MED1, CDK9 and RNA PolII-S2P/S5P, a complex highly competent for gene transcriptional activation. Importantly, our findings provide mechanistic details to a model in which BRD4-NUT recruitment of p300 through bipartite NUT TAD interaction with p300 TAZ2 domain triggers robust p300 HAT activation and resulting enrichment of H3K27ac in chromatin (Fig. 7f), which in turn recruits more BRD4-NUT and p300, nucleating a feed-forward loop that propagates a wide spread of BRD4-NUT/p300 nuclear condensation-assisted unusually large topological domain structure (megadomain) in chromatin. The BRD4-NUT/p300 megadomain structure is likely confined by factors including major topologically associated domains (TADs) and competition between limited HATs and HDACs in large excess in the neleus[51]. Such large-scale alternations of chromatin landscape can result in aberrant gene transcription by sequestration of limited p300 to the highly concentrated transcriptional hub that favors activation of anti-differentiation genes for perpetual tumor cell growth in NC carcinoma.

NC is a highly aggressive solid tumor with limited treatment options other than surgery and chemotherapy[9,12,16]. BET BrD inhibitors

have been evaluated in human clinical studies and shown initial efficacy for NC treatment[52–55]. However, NC patients develop drug resistance and relapse during the treatment[56,57], prompting an urgent need for new therapeutic strategies. Our mechanistic insights from this study suggest directions in drug development targeting BRD4-NUT/p300. We showed that spatially constrained inhibition of BRD4 tandem BrDs by bivalent inhibitor MS645[40] is much more effective than monovalent inhibitor JQ1 and HDAC inhibitors in disrupting BRD4-NUT/p300 nuclear condensation in chromatin. In addition, given the role of the BRD4-NUT TAD/TAZ2 binding in p300 recruitment and activation, chemical inhibition of the TAZ2 domain may be exploited to block BRD4-NUT/p300 interaction and to inhibit p300 activation and histone hyperacetylation required for tumorigenesis of NCs.

Pathological findings indicate that NCs resemble those of squamous cancer progression[41,47], but how BRD4-NUT fusion protein drives NC invasion and metastasis is insufficiently understood. ALX1 is a homeodomain-containing transcription factor, and recently implicated as a regulator of Snail expression in activating cancer invasion and migration[43–45]. In NCs, excessive histone hyperacetylation domains enclose the *ALX1* gene region[14], and possibly change its expression as well as cellular functions controlled by ALX1. In this study, we showed that BRD4-NUT occupies at the promoter of *ALX1*, promotes co-enrichment of p300, H3K27ac and BRD4L at its loci and activation of its transcriptional expression. Overexpression of BRD4-NUT is causally correlated with the upregulation of signature genes that favor the EMT signaling pathway and tumorigenesis of NC, whereas depletion of BRD4-NUT/p300 binding reduces *ALX1* expression and consequently results in EMT signaling suppression. Our study suggests that the BRD4-NUT/p300 bipartite interaction likely plays an important role for propagating histone hyperacetylation, chromatin condensation and sustaining aberrant gene transcription to perpetual tumor cell growth in NC carcinoma.

## Methods

### Recombinant plasmids construction

Fragment of p300-TAZ2 (aa 1723–1812) was amplified by PCR using pcDNA3.1-p300 (Addgene, #23252) as a template and sub-cloned into a pMAL-c5x vector (NEB, N8108S). TEV cleavage site was then sub-cloned between MBP and the N terminus of p300-TAZ2. All cysteines, which do not participate in zinc coordination, were mutated to alanines (C1738, C1746, C1769, C1770) by Quikchange Site-Directed Mutagenesis Kit (Stratagene, #200518). Fragment of NUT (aa 419–470) was amplified and sub-cloned into the C terminus of p300-TAZ2 construct separated by a GS5-linker. The coding sequence of cherry was amplified from pCherry-C1 construct and sub-cloned into the C terminus of p300-TAZ2 construct with a GS5-linker.

GST fusion constructs encoding full-length NUT and GFP-BRD4-NUT were kindly provided by Dr. Jianxin You. NUT fragments (aa 300–700, 346–592, 346–480, 470–592, 346–376, 377–418, 419–470) were generated by cloning the PCR-amplified coding fragments into the pGEX-4T-1 vector. GFP-BRD4-NUT-ΔF1c (fragment of NUT (aa 346–470) was deleted) construct was generated by overlapping PCR using QuikChange Site-Directed Mutagenesis Kit (Stratagene, #200518). NUT (aa 346–592) and (aa 300–700) fragments were amplified using the appropriate sets of primers and sub-cloned into pETHM-GFP construct which is previously described[39]. All plasmids were verified by DNA sequencing.

### Protein purification

TAZ2, TAZ2-NUT fusion protein and TAZ2-cherry protein expression were performed in BL21 (DE3) pLysS cells in TB medium supplemented with 100 μM ZnCl2, 2 g/L glucose for 16 h at 22 °C. Labeled protein expression was performed in M9 medium with the addition of 1 g/L of $^{15}$NH4Cl or 2 g/L $^{13}$C-glucose. Purification was performed as previously described[29].

### GST-NUT fragments

Protein expression was performed in BL21 (DE3) pLysS cells in TB medium for 16 h at 16 °C. Cells were collected and resuspended in cold lysis buffer (1× PBS, 1 mM DTT, pH 7.4) under addition of protease inhibitor (Roche). After sonication and cell debris removal by high-speed centrifugation (23,447 × g), the cell lysate was loaded onto GST column. The protein was eluted with elution buffer (50 mM Tris-HCl, 10 mM reduced glutathione, pH 8.0). Then, the protein was purified by size exclusion chromatography using Superdex 75 or 200 columns (GE Healthcare). The protein was concentrated by centrifugation (2240 × g) (Amicon Centrifugal Filter 10 kDa, MWCO), and concentrations were measured by UV absorbance at 280 nm and using the calculated extinction coefficient. The size and purity of the purified proteins were monitored by SDS-PAGE.

### Pull-down experiments

For recombinant protein pull down assays, 10–20 μg of each purified protein in pull-down buffer (1 × PBS, 1 mM DTT, 1 mM PMSF, pH 7.4) was combined and incubated with 25 μL GST affinity resin for 1 h at 4 °C under constant mixing. The resin was washed for four times with the same buffer and the complexes were eluted off the beads using elution buffer (50 mM Tris-HCl, 10 mM reduced glutathione, pH 8.0). The elutes were clarified by centrifugation 10 min 18,800 × g, supernatant and input samples were collected for western blot. GST fusion proteins were separated by SDS-PAGE and visualized using Coomassie blue staining.

For cell lysate GST pull-down assay, HEK293T cells were transfected with pcDNA3.1-p300, and cells were collected after 48 h. 500 μg GST or GST-NUT (aa 300–700) protein was incubated with 50 μL GST beads at 4 °C for 3 h in binding buffer (1×PBS, pH 7.4, 1 mM DTT, 1 mM PMSF). The beads were washed four times with binding buffer and incubated with the HEK293T cell lysates at 4 °C overnight. The beads were washed four times in binding buffer and eluted with binding buffer containing 10 mM reduced glutathione. Samples were resolved on SDS-PAGE and processed for Western blotting.

### NMR spectroscopy

All $^{15}$N/$^{13}$C labeled proteins samples were measured in 200 mM phosphate buffer pH 6.3 containing 10% D$_2$O or 100% D$_2$O.

### Chemical shift perturbations and K$_D$ measurement

Chemical shift perturbation experiments were performed at 50 μM protein concentration and the same buffer and pH conditions as above. Each fragment was titrated in each protein at increasing concentration in a range between 7.8 μM and 1000 μM. As reference spectrum was used the free form. The weighted chemical shift difference ($\Delta\delta_{weighted}$) was calculated using the equation: $\Delta\delta_{weighted} = \sqrt{|\Delta\delta H|^2 + |\Delta\delta N|^2 * 0.15}$, where $\Delta\delta H$ is the chemical shift on the proton and $\Delta\delta N$ is the chemical shift on the nitrogen which is scaled with a factor 0.15 to account for the different range of the amide proton and amide nitrogen. Chemical shifts for each backbone amide group were measured from the peak detected in apo form spectrum to the peak at the end of the titration.

K$_D$ of binding for each fragment were estimated using the following equation:

$$\Delta\delta obs = \Delta\delta\,max \frac{\{Kd + Pt + Lt - \sqrt{(Kd + Pt + Lt)^2 - 4Pt*Lt}\}}{2Pt} \quad (1)$$

[P]t and [L]t are the total concentration of protein and ligand; Δδobs is the observed shift in regard to the reference, while Δδmax is the maximum shift obtained upon saturation and is extracted from the fitting. K$_D$ for each fragment was extrapolated as an average value of KD +/− S.E.M. of those resonances.

## NMR data collection

The p300 TAZ2 domain/NUT peptide complex and TAZ2-NUT fusion protein were used for structure determination. NMR samples of the TAZ2 domain (0.5 mM) in complex with a NUT peptide (aa 403–418, Supplementary Table 1) of 1.5 mM and TAZ2-NUT fusion protein (0.5 mM) were prepared in 200 mM phosphate buffer (pH 6.3) in H$_2$O/D$_2$O (9/1) or 100% D$_2$O. All nuclear magnetic resonance (NMR) spectra were acquired at 25 °C on Bruker 600, 800 MHz spectrometers equipped with Z-gradient triple-resonance cryoprobes (Bruker Top-Spin v3.0). The backbone $^1$H, $^{13}$C, and $^{15}$N resonances were assigned using standard three-dimensional triple-resonance HNCA, HN(CO)CA, HN(CA)CB, and HN(COCA)CB experiments. The side-chain atoms were assigned from three-dimensional HCCH-TOCSY, HCCH-COSY, and (H)C(CO)NH-TOCSY data. The NOE derived distance restraints were obtained from $^{15}$N- or $^{13}$C-edited three-dimensional NOESY spectra. The NUT peptide was assigned from two-dimensional TOCSY, NOESY, ROESY, and $^{13}$C/$^{15}$N-filtered TOCSY and NOESY. The intermolecular NOEs used in defining the structure of the complex were detected in $^{13}$C-edited (F1), $^{13}$C/$^{15}$N-filtered (F3) three-dimensional NOESY spectra (unlabeled NUT peptide bound to $^{13}$C/$^{15}$N-labeled TAZ2 protein). Spectra were processed with NMRPipe (v2.0)[58] and analyzed using NMRVIEW (v5.0)[59].

## Structure calculations

Structures of the p300 TAZ2 domain/NUT peptide (aa 403–418) and TAZ2-NUT fusion protein were calculated with a distance-geometry simulated annealing protocol with CNS[60]. Initial protein structure calculations were performed with manually assigned NOE-derived distance constraints. hydrogen-bond distance, φ and ψ dihedral-angle restraints from the TALOS + (v3.70F1) prediction were added at later stage of structure calculations for residues with characteristic NOE patterns[61]. The converged structures were used for the iterative automated NOE assignment by ARIA refinement[62]. Structure quality was assessed with CNS (v1.5), ARIA (v2.0), and PROCHECK (v.3.5.4) analysis [60,62,63]. A family of 200 structures was generated and 20 structures with the lowest energies were selected for the final analysis.

## Cell culture and transfection

The human embryonic kidney cell line HEk293T and human normal liver cell line LO2 were cultured in DMEM (with glutamine) containing 10% FBS (BI) and 100 µg/mL Penicillin/Streptomycin. Polyethyleneimine (Polysciences) transfection reagent was used for transient transfection in all of these cells.

## Cell lysis and western blotting

Cells were lysed with cell lysis buffer (Cell Signaling Tech, #9803) containing Protease Inhibitor Cocktail (Transgene, DI101-01) and PhosSTOP (Roche, 4906837001). Total protein concentration was determined by BCA assay using copper (II) solution (Sigma Aldrich, C2284) and bicinchoninic acid solution (Sigma B9643). The cell extracts were mixed with 6× SDS-PAGE sample buffer (300 mM Tris-HCl, pH 6.8, 0.12 g/mL SDS, 100 mM DTT, 60% glycerol, 0.6 mg/mL bromophenol blue) and heated at 95 °C for 10 min. 50 µg total protein of each sample was loaded into SDS-PAGE. Then, samples were transferred to a PVDF membrane. The membrane was blocked in 5% skim milk and incubated with indicated antibodies overnight at 4 °C. Detection was performed using an HRP goat anti-mouse/rabbit IgG peroxide conjugate. The following antibodies were used: Rabbit Monoclonal anti-GAPDH (Abcam, Cambridge, UK) (#ab181602, 1:5000); Rabbit Monoclonal anti-Histone H3 (acetyl K18) (Abcam, Cambridge, UK) (#ab40888, 1:10000); Rabbit Monoclonal anti-Histone H3 (acetyl K27) (Abcam, Cambridge, UK) (#ab4729, 1:2000); Rabbit Monoclonal anti-Histone H3 (acetyl K56) (Abcam, Cambridge, UK) (#ab71956, 1:3000); Rabbit Monoclonal anti-Histone H3 (Abcam, Cambridge, UK) (#ab1791, 1:2000); Rabbit Monoclonal anti-KAT3B/

p300 (Abcam, Cambridge, UK) (#ab54984, 1:2000); Mouse Monoclonal anti-p53 (Abcam, Cambridge, UK) (#ab1101, 1:2000); Goat Polyclonal anti-SNAIL (Abcam, Cambridge, UK) (#ab53519, 1:2000); Rabbit Monoclonal anti-Alx1 (Abcam, Cambridge, UK) (#ab181101, 1:2000); Acetyl-Histone H4 (Lys12) (D2W6O) Rabbit mAb (Cell Signaling Tech, Danvers, MA) (#13944, 1:1000); Acetylated-Lysine Antibody, Rabbit, (Cell Signaling Tech, Danvers, MA) (#9441, 1:1000); E-Cadherin (4A2) Mouse mAb (Cell Signaling Tech, Danvers, MA) (#14472, 1:1000); Vimentin (D21H3) XP® Rabbit mAb (Cell Signaling Tech, Danvers, MA) (#5741, 1:1000); N-Cadherin (D4R1H) XP® Rabbit mAb (Cell Signaling Tech, Danvers, MA) (#13116, 1:1000); Rabbit Polyclonal Anti-acetyl-p53 Antibody (Lys373, Lys382) (Merck Millipore, Burlington, MA) (#06-758, 1:2000); Monoclonal Anti-Involucrin antibody produced in mouse (Sigma-Aldrich, Burlington, MA) (#i9018, 1:1000); Anti-GFP tag Mouse mAb (Engibody, Dover, DE) (#AT0028, 1:2000); GFP-tag Antibody Mouse mAb (Affinity, Cincinnati, OH) (#T0005, 1:2000); Mouse Monoclonal anti-MBP tag (Proteintech, Rosemont, IL) (#66003, 1:2000); Goat Anti-Rabbit IgG H&L (HRP) (Abcam, Cambridge, UK) (#ab6721, 1:10000); Goat Anti-Mouse IgG H&L (HRP) (Abcam, Cambridge, UK) (#ab205719, 1:10000), Rabbit Anti-Goat IgG H&L (HRP) (Abcam, Cambridge, UK) (#ab6741, 1:5000).

## RNA isolation, reverse transcription, and quantitative PCR

The total RNA was isolated and purified using the Trizol Reagent. 1 µg of total RNA was reverse-transcribed using the TransScript First-Strand Synthesis SuperMix Kit (TransGene, AT351). qPCR was run on a LightCycler® 480 instrument (Roche) using Fast SYBR Green Master Mix (Applied Biosystems, Cat#: 4385617). mRNA levels were determined by the relative standard curve method, normalized to GAPDH and presented as relative mRNA levels. qPCR analyses were done in triplicate (LightCycler 480 Software v1.5.1.62). Experiments were repeated at least twice. Primers are listed in Supplementary Table 3.

## In vitro droplet assays

In vitro droplet assays were performed as previously described with slightly change[39]. Recombinant 6×His-EGFP-NUT (aa 300–700) was stored in 1×PBS, pH 7.4, and was supplemented with 360 mM NaCl. Recombinant 6× His-Cherry TAZ2 was stored in 25 mM Tris-HCl, pH 6.3, 200 mM NaCl, 25 µM ZnCl$_2$. Protein was diluted in 1×PBS, pH7.4 buffer supplemented with indicated NaCl concentrations and 10% PEG-6000. The protein solution was loaded either into a 384-well plate or a homemade chamber comprising a glass slide with a coverslip attached by two parallel strips of tape.

## Immunofluorescence staining

Cells were cultured on coverslips and fixed with 4% PFA for 10 min at room temperature. Cells were incubated in permeabilization buffer (0.1% Triton X-100) in PBS for 20 min at room temperature and blocked (3% BSA, 2% goat serum in PBS) for 60 min at room temperature. The following primary antibodies were used: Rabbit Monoclonal anti-Histone H3 (acetyl K18) (Abcam, Cambridge, UK) (#ab40888, 1:1000); Rabbit Monoclonal anti-Histone H3 (acetyl K27) (Abcam, Cambridge, UK) (#ab4729, 1:2000); Rabbit Polyclonal anti-TRAP220/MED1 (Abcam, Cambridge, UK) (#ab64965, 1:500); Rabbit Polyclonal anti-CDK9 (Abcam, Cambridge, UK) (#ab6544, 1:600); Rabbit Polyclonal anti-RNA polymerase II CTD repeats YSPTSPS (phospho S5) (Abcam, Cambridge, UK) (#ab5131, 1:1000); Rabbit Polyclonal anti-RNA polymerase II CTD repeats YSPTSPS (phospho S2) (Abcam, Cambridge, UK) (#ab5095, 1:500); Mouse monoclonal ANTI-FLAG® M2 antibody (Merck Millipore, Burlington, MA) (#F1804, 1:1000). After staining overnight at 4 °C, cells were washed by permeabilization buffer three times. Secondary antibodies used were: Goat anti-Mouse IgG (H + L) Cross-Adsorbed Secondary Antibody, Alexa Fluor™ 594 (Invitrogen, Carlsbad, CA) (#A-11005, 1:1000); Goat anti-Rabbit IgG (H + L) Cross-Adsorbed Secondary Antibody, Alexa

Fluor™ 594 (Invitrogen, Carlsbad, CA) (#A-11012, 1:1000). Cells were counter-stained with 4,6-diamidino-2-phenylindole (DAPI) and examined under an Olympus Confocal microscope (FV3000) (OLYMPUS cellSens Standard v1.18).

## Fluorescence recovery after photo-bleaching (FRAP) imaging

FRAP was performed on an upright Zeiss confocal microscope. Images were acquired using a ×63 oil-immersion objective at a zoom corresponding to $100 \times 100$-$nm^2$ pixel size, and the microscope was controlled using Zeiss Zen software. In most FRAP experiments, except where otherwise noted, 50 frames were acquired at 1 frame per second, allowing 3 frames to be acquired before the bleach pulse. A circular bleach spot was chosen in a region of homogenous fluorescence at a position at least 1 μm from the nucleus. The spot was bleached using maximal laser intensity and pixel dwell time corresponding to a total bleach time of -1 s.

Image analysis was performed using Image J software (ZEN 2012, Blue Edition). FRAP profiles were calculated using a ROI marked at the bleached area and use the plug-in FRAP profiler to obtain fluorescence intensity profiles. Fluorescence intensities in the ROI immediately after bleaching $F(0)$ were subtracted from fluorescence intensities at all times $F(t)$ and results were then normalized to pre-bleaching values. Results were then imported into Prism software for statistics analysis. We generally collected data from 3 cells per cell line per condition per day, and all presented data are from at least three independent replicates on different days. All data were pooled and fluorescence intensity at time points after the bleaching step were fitted to the equation:

$$\text{Fluorescence Recovery} = \frac{F(t) - F(0)}{F(-t) - F(0)} = Mf \cdot (1 - e^{\frac{ln2 \cdot t}{t_{1/2}}}) \qquad (2)$$

where $F(t)$, $F(-t)$ and $F(0)$ are the average fluorescence of the ROI at any time, before bleaching and, immediately after bleaching, respectively. $Mf$ is the mobile fraction, $t_{1/2}$ is the half time of recovery and t is time in seconds. In Prism, this fitting is achieved by using non-linear regression and the exponential one-phase association model using $Y_0 = 0$ and where $Mf$ corresponds to the plateau value. We note that because the bleach duration was relatively long compared with the timescale of molecular diffusion, it is not possible to accurately estimate the immobile and free fractions from our FRAP curves.

## HEK293T cell nuclear extract preparation

HEK293T cells were transfected with pEGFP-C3, pEGFP-BRD4-NUT, pEGFP-BRD4-NUT-ΔF1c, pEGFP-BRD4-NUT-ΔTAD1, pEGFP-BRD4-NUT-ΔTAD2, pEGFP-p53, pEGFP-p53-ΔTAD1, pEGFP-p53-ΔTAD2, or pEGFP-p53-ΔTAD1/ΔTAD2 for 48 h. HEK293T cell nuclear extract was prepared by the previously described method with slight modification[64]. Cells were sedimented at $1000 \times g$ for 5 min at 4 °C and washed with ice-cold PBS. Pelleted cells were resuspended in 5 volumes of DR Buffer A (10 mM HEPES-KOH, pH 7.9, 10 mM KCl, 1.5 mM $MgCl_2$, 0.5 mM DTT, 0.5 mM PMSF) and incubated on ice for 15 min. Cell suspensions were lysed with 35 strokes of a Dounce tight pestle (Wheaton), and nuclei were pelleted at 9600xg for 20 min, at 4 °C. Supernatants were decanted and the resulting pellet was gently resuspended in DR Buffer C (0.2 mM EDTA, 25% glycerol, 20 mM HEPES-KOH, pH 7.9, 420 mM NaCl, 1.5 mM $MgCl_2$, 0.5 mM DTT, 0.5 mM PMSF) at 3 ml per $10^9$ cells and rotated for 60 min at 4 °C. The suspension was centrifuged at 10,000 rpm for 30 min at 4 °C, and supernatants were collected as the nuclear fraction and underwent buffer change to HAT reaction buffer (50 mM Tris-HCl, pH 8.0, 0.1 mM EDTA, 50 mM NaCl, 1 mM DTT, 10% glycerol, 1 mM PMSF, 10 mM sodium butyrate supplemented with protease inhibitors) and were concentrated by centrifugation (2240 × $g$) using Amicon Ultra centrifugal filters (3 K MWCO, Millipore).

## Co-immunoprecipitation

HEK293T cells transfected with indicated plasmids for 48 h were lysed in the lysis buffer (20 mM HEPES, pH 7.9, 300 mM NaCl, 3 mM $MgCl_2$, 0.1% NP-040) containing Protease Inhibitor Cocktail (Transgene, DI101-01) and 10 mM sodium butyrate for 30 min at 4 °C with rotation. After a 20 min centrifugation at 18,800 × $g$ at 4 °C, the supernatants were collected and separated into input and immunoprecipitated (IP) lysates. IP lysates were mixed with a GFP specific antibody overnight at 4 °C, and then were immunoprecipitated with 25 μl Dynabeads™ Protein G (pre-washed with PBS buffer three times, Invitrogen) for 4 h at 4 °C. After washing with the same IP lysis buffer, IPs were released by adding 2× Protein Loading Buffer (TransGen Biotech) to beads and subjected to western blotting.

## HAT assay

HAT assays were performed in HAT reaction buffer (50 mM Tris-HCl, pH 8.0, 0.1 mM EDTA, 50 mM NaCl, 1 mM DTT, 10% glycerol, 1 mM PMSF, 10 mM sodium butyrate supplemented with protease inhibitors) containing 1 μg histone H3 (NEB), 100 μM acetyl-CoA and 40 μg HEK293T cell nuclear extracts (80 μg for H3K56ac). The reaction was incubated at 30 °C for 1 h and stopped by adding 1× Protein Loading Buffer (TransGen Biotech) and boiling for 10 min. Samples were resolved on SDS-PAGE and processed for Western blotting.

## ChIP assay

For ChIP experiments, cells were cross-linked with formaldehyde before chromatin was extracted, sonicated, and incubated with primary antibodies Rabbit Monoclonal anti-Histone H3 (acetyl K27) (Abcam, Cambridge, UK) (#ab4729, 1:500); Mouse monoclonal ANTI-FLAG® M2 antibody (Merck Millipore, Burlington, MA) (#F1804, 1:100); Rabbit Polyclonal anti-BRD4 (Bethyl, Montgomery, TX) (#A301-985A100, 1:100); Mouse Monoclonal anti-KAT3B/p300 (Abcam, Cambridge, UK) (#ab14984, 1:100); Normal mouse IgG (Santa Cruz, Santa Cruz, CA) (#sc-2025, 1:200); Normal rabbit IgG (Cell Signaling Tech, Danvers, MA) (#2729, 1:200) overnight. Antibody complexes were then captured with Dynabeads Protein G (Thermo Fisher, 10004D), and DNA was eluted, de-crosslinked, and purified. ChIP signal were calculated by qPCR (Supplementary Table 4) relative to input levels after (IgG) background subtraction.

## In vitro wound healing assay

Transfected cells were seeded on 6-well plates until grown to 80% confluency. The transfection efficiency is detected by the positive EGFP signal, all the transfection efficiency is over 90%. The protein expression level is also analysis by western blot and result is shown in Fig. 4e. Cells were scratched by use of a 200 μL pipette tip. After washed three times with PBS to remove debris, cells were cultured in FBS-free medium to block cell growth. Recovery from the scratched wound was monitored by the use of a microscope. Each migration experiment was carried out in triplicate and repeated for three times. The relative migration rate was calculated using ImageJ software (ImageJ 1.48 V). Distance migrated was calculated by subtracting the average distance between wound edges from that at the beginning.

## Gene knockdown by siRNA

NC 14169 cells were seeded to a density of $2 \times 10^5$ cells/well in a six-well plate. siRNA mixtures against *Myc*, *Alx1* and *Nutm1* were obtained from Dharmacon and prepared by diluting siRNA in serum-free transfection medium and mixed with DharmaFECT#1 diluted in serum-free transfection medium. The NC 14169 cells were then treated with the siRNA mixtures for 24 and 72 h, and cells were collected for qPCR and western blot analysis of mRNA and protein expression levels, respectively.

## Analytical size exclusion chromatography

Purified MBP-TAZ2, His-F1c (aa 346–592) or MBP-TAZ2/His-F1c complexes (0.8 mM) were injected and eluted from the column (GE, Superdex 200 increase 10/300 GL) at a flow rate of 0.3 ml/min. The column was equilibrated with PBS buffer and calibrated using the Gel Filtration Markers Kit (Sigma, MWGF1000) comprising a set of molecular weight protein standards in the same buffer condition as indicated.

## Droplet pelleting assay

EGFP-NUT (aa 300–700, 10 µM) and TAZ2 (molar ratio 1:2) mixtures were subjected to in-vitro droplet assay. The complex samples were spun at $2500 \times g$ for 5 min after post-incubation time as indicated. The supernatants (S) were collected and pellets (P) resuspended in PBS buffer. Both supernatants and pellets fractions were prepared in 4×Laemmli Sample Buffer (Biorad, #1610747) and heated to 60 °C for 5 min prior to running SDS-PAGE. The resolved proteins were transferred to 0.22 µm PVDF membrane, immunoblotted with specific antibody, and revealed by an enhanced chemiluminescence method.

## Protein sequence analysis

The disorder of NUT protein sequence was predicted using an online Predictor of Natural Disorder Regions (PONDR VSL2, http://www.pondr.com/)[33]. Secondary structures of the NUT protein sequence were predicted using an online Partition and Semi-Random Subspace Method (PSRSM, http://qilubio.qlu.edu.cn:82/protein_PSRSM/default.aspx)[34].

## Statistics and reproducibility

Experimental data are presented as mean +/− S.E.M or S.D. Sample numbers are indicated in the relevant figure legends. All in vitro and cellular experiments were independently replicated at least twice and repeated at least three times within each of the experimental runs. Statistical significance calculations comparing two conditions were performed using a two-tailed unpaired Student's t-test (GraphPad Prism v6.0c and Excel for Mac v14.3.0) or two-sided $p$-values for each indicated comparison are derived from the R software loaded with Limma package (R software v3.2.3, Limma package v3.26.8), with $p < 0.05$ considered statistically significant unless stated otherwise. No statistical methods were used to predetermine sample size.

## Reporting summary

Further information on research design is available in the Nature Portfolio Reporting Summary linked to this article.

# Data availability

The data that support this study are available from the corresponding authors upon reasonable request. The Structure coordinate data for the p300 TAZ2 domain in complex with NUT TADs peptides generated in this study have been deposited in the Protein Data Bank database under PDB accession code 7XEZ and 7XFG. The NMR data are deposited in the Biological Magnetic Resonance Data Bank (BMRB code 36480 and 36481). Source data are provided with this paper.

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

## Acknowledgements

We thank Prof. Jianxin You at University of Pennsylvania Perelman School of Medicine for providing valuable NUT constructs, and Prof. Christopher A. French at Harvard Medical School for NC cell lines, and L. Sun for assistance with microscopy. We thank the State Key Laboratory of Supramolecular Structure and Materials at Jilin University for the use of their research facilities. This work was supported in part by the research fund from the First Hospital of Jilin University (Changchun, China), the Open Project of State Key Laboratory for Supramolecular Structure and Materials, JLU (SKLSSM201602), JLU Science and Technology Innovative Research Team (JLUSTIRT, 2017TD-25), International Center of Future

Science, JLU, and National Natural Science Foundation of China (31770780; L.Z.).

## Author contributions

D.Y., Y.L., Y.J., and R.G. performed BRD4-NUT transfection and microscopic imaging studies in cells, X.H. and X.Y. NUT pulldown and condensation experiments, C.K., A.J., and K.L.C. NMC cellular experiments, D.Y., C.W., and L.Z. NMR experiments and protein structure calculations, Y.J., X.Y., D.J., and Q.Z. protein expression and purification. L.Z. and M.-M.Z. wrote the manuscript with input from all co-authors.

## Competing interests

M.-M.Z. is a scientific founder, director and shareholder of Parkside Scientific Inc. Remaining authors do not declare any competing interests.

## Additional information

[1]Bethune Institute of Epigenetic Medicine, The First Hospital of Jilin University, Changchun, Jilin 130021, China. [2]International Center of Future Science, Jilin University, Changchun 130012, China. [3]Department of Pharmacological Sciences, Icahn School of Medicine at Mount Sinai, New York, NY 10029, USA. [4]State Key Laboratory of Supramolecular Structure and Materials, College of Chemistry, Jilin University, Changchun, Jilin 130012, China. [5]These authors contributed equally: Di Yu, Yingying Liang. ✉e-mail: ming-ming.zhou@mssm.edu; leizeng@jlu.edu.cn

