## [Peer Review File · Nature Communications]

REVIEWER COMMENTS

Reviewer #1 (Remarks to the Author):

In this manuscript, Yu and colleagues investigate how the NUT part of the oncogenic BRD4-NUT fusion binds to the TAZ2 domain of p300 and what is the biological consequence of this interaction. The authors report two NMR structures – the TAZ2 domain in complex with two different regions of NUT – and carefully examine the formation of both complexes by IPs, NMR and FPLC. The authors show that BRD4-NUT forms phase separated condensates in vitro and in vivo and that binding of p300 promotes phase separation. The authors further demonstrate that BRD4-NUT/p300 containing condensates drive the activation of transcription of pro-proliferative and oncogenic genes. From my point of view, this is a very well executed study- it contains excellent quality data, and conclusions are convincing and justified. Importantly, this study provides mechanistic details about proteins directly implicated in NUT carcinoma which might pave the way for the development of novel therapeutics against this devastating aggressive form of cancer. This work can be of high interest not only to scientists working in the field of chromatin biology but to a wide scientific community, and I enthusiastically support publication.

A couple of minor suggestions- to generate a couple of mutants to disrupt the NUT-TAZ2 interphase (focusing on disease-related mutations particularly, if any are known), and to measure the HAT activity of p300 in the presence of BRD4-NUT.

Reviewer #2 (Remarks to the Author):

In this study, the authors investigated the molecular mechanism through which the fusion protein BRD4-NUT recruits and activates the histone acetyltransferase (HAT) p300 and subsequently assessed the functional bearing of this interaction on the expression of pro- and anti-proliferation/differentiation gene sets and signaling pathways. Overall, the objective of the study is clearly stated and the research question is timely and interesting, the methods are appropriate, the results are convincing and the discussion is well argued. I only have few concerns that need to be addressed by the authors:

1. It is not clear whether the transactivation domains (TADs) in BRD4-NUT targeting and activating TAZ2 domain is P300-specific or is it more universal. In other words, could TDAs in BRD4-NUT target and activate other TAZ2 domain-containing proteins? The authors could address this question by P300 knockdown type of work.
2. The clear data in Figs. 1-3 and the observation that BRD4-NUT fusion is a driver of P300 recruitment and activation notwithstanding, the capacity of TADs in unfused NUT to target TAZ2 domains in P300, activate P300 as a HAT to induce hyperacetylation in target cells was not addressed.
3. In Fig. 5, the authors stated that BRD4-NUT fusion protein up-regulates ALX1 and promotes EMT, however, the data supporting this claim is thin (pPCR of key genes like SNAIL); there is no data on loss of cell polarity, cell-cell adhesion, gain of migratory and invasive properties, etc. to show that changes in such EMT-related gene expression translates into factual EMT.
4. The functional consequences of BRD4-NUT/P300 interaction and the resulting hyperacetylation need to be further investigated; additional data points (e.g. resistance to targeted chemotherapy [MS645, JQ1], anti-apoptotic potential, invasion potential, cell cycling [CDK9], cancer stem-cell formation potential) and further discussed is required.
5. Is the consequent hyperacetylation restricted to H3K27 as shown in Fig. 4c; could other residues (e.g. H3K18) be targeted?
6. In the diagrammatic representation (Fig. 6d), it is not clear why the authors focused on AXL1 and other anti-differentiation genes despite the fact that data in Fig. 6 shows that BRD4-NUT/P300 interaction alters the expression of other gene sets such as the EMT-related genes.

Reviewer #3 (Remarks to the Author):

It is the first time that the structure of the p300 BRD4-NUT/p300 complex is resolved, providing a novel drug target to block BRD4-NUT/p300 interaction to treat this devastating disease. Besides this, this manuscript mainly confirms the Reynoird et al.'s findings in 2010. In Reynoird's paper, the authors identified p300 as a binding partner of BRD4-NUT through interaction between p300 TAZ2 domain and BRD4-NUT F1c region. This interaction can activate p300, leading to enhanced HAT activities of p300. Although this manuscript further explored one of the downstream targets ALX1 in the tumorigenesis of NUT carcinoma (NC), most cellular experiments were done in Non-NC models.

Concerning the manuscript itself, I have the following issues that the authors could address.

1. I think the oncogenesis of NC relies on the whole aberrant gene transcription landscape, and ALX1 is within this landscape. Although the authors showed EMT induction by wt BRD4-NUT but not by BRD4-NUT with F1c deletion in non-NC models, the direct evidences whether ALX1 is involved are still missing. The authors could simply check if EMT induction by wt BRD4-NUT is compromised by ALX1 depletion. To prove its vital role in NC, the biological functions of ALX1 also needs to be specifically addressed in NC models. For example, ALX1 knockdown experiments could be performed in NC to test its functions in controlling cell proliferation, squamous differentiation and EMT.
2. The overexpression of BRD4-NUT with F1c deletion still forms foci and colocalized with histone hyperacetylated foci (Fig. 4c & d), though to a less extent compared to wt BRD4-NUT. I am wondering if other regions of BRD4-NUT still recruits other HATs. The authors have many BRD4-NUT deletion constructs and it would be interesting to test them in the same setting.
3. In Figure 4f, the authors observed enhanced H3K27ac, but not H3K56ac and H4K12ac in BRD4-NUT overexpressed cell lysates. But in Figure 4g, the in vitro HAT assay showed both enhanced H3K27ac and H3K56ac from the same nuclear extracts. Can the authors explain the discrepancy?

Point-to-Point Response to Reviewers' Comments (NCOMMS-22-25872-T)

We thank all three Reviewers for their constructive comments and helpful suggestions. In this revision, we have fully addressed the reviewers' comments through clarifications and by providing additional new data that have improved the presentation and strengthened the conclusions of our study. Below, we list the major changes in the revision, followed by our point-to-point responses to the reviewers' comments.

A. Major Changes:

Figure 4

Fig. 4a, b – We provided new data to assess the effects of single TAD1 or TAD2 deletion mutation EGFP-BRD4-NUT- Δ TAD1 or Δ TAD2 on co-condensation with p300 in LO2 cell nucleus.

Fig. 4c – We provided new data showing H3K18ac co-condensation with wild-type EGFP-BRD4-NUT-WT or the TAD1/2 deletion mutant EGFP-BRD4-NUT- Δ F1c in LO2 cell nucleus.

Figure 5 (previous Figure 4f, g)

Fig. 5a – We updated the western blot analysis showing that the effects of ectopic expression of EGFP-BRD4-NUT-WT or TAD1/2 deletion mutants (Δ TAD1, Δ TAD2, and Δ F1c) in LO2 cells on p300 acetylation, activation and histone acetylation.

Fig. 5b – We updated the results of the *in-vitro* acetylation assay with HEK293 nuclear extracts, over-expressing EV, EGFP-BRD4-NUT-WT or TAD1/2 deletion mutants.

Figure 7 (previous Figure 6)

Fig. 6d, e – We presented new data showing siRNA knockdown of MYC, ALX1, NUTM1 in NC 14169 cells, and their effects on the protein expression level of Involucrin, a differentiation marker.

Extended Data Figure 5

ED-Fig. 5b, c – We provided new data showing the effects of 12 hour- and 48 hour-treatment of BRD4 inhibitors JQ1 and MS645, or HDAC inhibitors on p300 co-condensation with EGFP-BRD4-NUT-WT in LO2 cell nucleus.

Extended Data Figure 6

ED-Fig. 6a – We provided new fluorescence imaging data showing Flag-BRD4L and Flag-BRD4S puncta co-condensation with EGFP-BRD4-NUT-WT or EGFP-BRD4-NUT- Δ F1c in LO2 cell nucleus.

Extended Data Figure 7

ED-Fig. 7a – We provided a new gel-shift assay data validating the BRD4-NUT WT and TAD1/2 deletion mutant plasmid DNAs used in this study.

ED-Fig. 7b, c – We provided new Co-IP data showing p53-WT interacted with endogenous p300 more than the TAD1/2 deletion mutants of p53, including p53- Δ TAD1, Δ TAD2 and Δ TAD1/2.

ED-Fig. 7d – We provided new western blot data showing that ectopic expression of wild-type EGFP-p53-WT elevated p300 autoacetylation and histone acetylation, whereas expression of the TAD deletion mutants (Δ TAD1, Δ TAD2 and Δ TAD1/2) only induced marginal enhancement.

ED-Fig. 7e – We generated new *in-vitro* acetylation assay data with HEK293 nuclear extracts, where we overexpressed EV, EGFP-p53-WT and TAD deletion mutants of p53- Δ TAD1, Δ TAD1 and Δ TAD1/2.

Extended Data Figure 8

We provided the analysis of available ChIP-seq and RNA-seq datasets from NC PER-403 and TC-797 cells for select BRD4-NUT target genes.

B. Reviewer #1 (Remarks to the Author):

In this manuscript, Yu and colleagues investigate how the NUT part of the oncogenic BRD4-NUT fusion binds to the TAZ2 domain of p300 and what is the biological consequence of this interaction. The authors report two NMR structures – the TAZ2 domain in complex with two different regions of NUT – and carefully examine the formation of both complexes by IPs, NMR and FPLC. The authors show that BRD4-NUT forms phase separated condensates in vitro and in vivo and that binding of p300 promotes phase separation. The authors further demonstrate that BRD4-NUT/p300 containing condensates drive the activation of transcription of pro- proliferative and oncogenic genes. From my point of view, this is a very well executed study- it contains excellent quality data, and conclusions are convincing and justified. Importantly, this study provides mechanistic details about proteins directly implicated in NUT carcinoma which might pave the way for the development of novel therapeutics against this devastating aggressive form of cancer. This work can be of high interest not only to scientists working in the field of chromatin biology but to a wide scientific community, and I enthusiastically support publication.

B. Q1. “A couple of minor suggestions- to generate a couple of mutants to disrupt the NUT-TAZ2 interphase (focusing on disease-related mutations particularly, if any are known), and to measure the HAT activity of p300 in the presence of BRD4-NUT.”

Response: We very much appreciate Reviewer #1’s positive appraisal of our study and constructive comments. BRD4-NUT is a translocation fusion protein and itself is responsible for NUT carcinoma (NC) and there are no other NC-related mutations reported for BRD4-NUT. To address the reviewer’s comment, and to further validate our findings that NUT-TAZ2 binding induces p300 HAT activation, we generated two single-TAD deletion mutants (BRD4-NUT- Δ TAD1, - Δ TAD2) (Fig. 4a). We found that the single deletion (Δ TAD1 or Δ TAD2) or double deletion (Δ TAD1/2) mutant showed nearly complete loss in their ability to activate p300 (Fig. 5a). This is in contrast to BRD4-NUT-WT, which activates p300 HAT activity as reflected by markedly increased levels of p300 acetylation (p300-Kac), and H3K18ac and H3K27ac in HEK293 cells. Further, in an *in-vitro* HAT assay with nuclear extracts of HEK293 cells overexpressing EGFP-BRD4-NUT-WT or the TAD deletion mutants, we confirmed that BRD4-NUT-WT increased acetylation levels of endogenous p300 (p300-Kac) and recombinant histone H3 (H3K27ac, H3K18ac, and H3K56ac), whereas neither of single nor double TAD1/2 deletion mutants yielded any significant enhancement of p300 or histone H3 acetylation as compared to the empty vector control (Fig. 5b). Taken together, our new data further highlight that BRD4-NUT’s bipartite interaction with p300 is required for p300 HAT *trans*-activation and histone H3 hyper-acetylation in chromatin.

C. Reviewer #2 (Remarks to the Author):

In this study, the authors investigated the molecular mechanism through which the fusion protein BRD4-NUT recruits and activates the histone acetyltransferase (HAT) p300 and subsequently assessed the functional bearing of this interaction on the expression of pro- and anti-proliferation/differentiation gene sets and signaling pathways. Overall, the objective of the study is clearly stated and the research question is timely and interesting, the methods are appropriate, the results are convincing and the discussion is well argued. I only have few concerns that need to be addressed by the authors:

C. Q1. “It is not clear whether the transactivation domains (TADs) in BRD4-NUT targeting and activating TAZ2 domain is P300-specific or is it more universal. In other words, could TDAs in BRD4-NUT target and activate other TAZ2 domain-containing proteins? The authors could address this question by P300 knockdown type of work.”

Response: We thank Reviewer #2 for his/her helpful feedback. The TAZ2 (transcriptional adaptor zinc finger) is a unique zinc finger domain only present in transcriptional coactivator HATs p300/CBP (Krois et al., 2016). The TAZ2 shares a similar topological fold as the TAZ1 domain, also present in p300/CBP, which consists of four α -helices stabilized by binding to three zinc atoms. However, the TAZ1 and TAZ2 differ significantly in the protein sequence and highly selective in interactions with their effector proteins. Indeed, it has been shown that the F1c domain (comprising TAD1/2) in BRD4-NUT selectively targets

the TAZ2 but not TAZ1 in p300 (Reynoird et al., 2010). Further, among many HATs examined, BRD4-NUT specifically binds to p300/CBP – but not PCAF, Tip60, HBO1 or HAT1 that do not contain TAZ2 – resulting histone hyper-acetylation in chromatin. These results highlight the selective interaction between BRD4-NUT TADs and p300-TAZ2.

Notably, many transcription factors such as p53 contain tandem transactivation domains (TADs) that interact with p300/CBP (Krois et al., 2016) and synergistically activate gene transcription (Candau et al., 1997). To investigate whether such tandem TADs form bipartite interactions with and activate p300, we generated p53 TAD deletion mutants (Δ TAD1, $-\Delta$ TAD2, or $-\Delta$ TAD1/2) (**ED-Fig. 7b**). We found that wild-type EGFP-p53-WT transfected in HEK293 cells interacts with p300, which was reduced with single or double TAD1/2 deletion mutations (**ED-Fig. 7c**), and that the EGFP-p53-WT, but not its TAD deletion mutants, increased levels of p300-Kac, H3K18ac, H3K27ac and p53-Kac in both cellular and *in-vitro* acetylation assays (**ED-Fig. 7d,e**). These results demonstrate that p53 TADs form bipartite interactions with p300, resulting in p300 and p53 activation and histone acetylation.

C. Q2. “The clear data in Figs. 1-3 and the observation that BRD4-NUT fusion is a driver of P300 recruitment and activation notwithstanding, the capacity of TADs in unfused NUT to target TAZ2 domains in P300, activate P300 as a HAT to induce hyperacetylation in target cells was not addressed.”

Response: We thank Reviewer #2 for the thoughtful comment. Previous studies show that expression of the unfused NUT is normally restricted to post-meiotic spermatogenic cells, where it binds to HAT p300 and/or CBP for enhancement of H4K5 and H4K8 acetylation and completion of the histone-to-protamine exchange (Shiota et al., 2018). In addition, *in-vitro* HAT study showed that the F1c domain of the unfused NUT is responsible for selectively targeting the TAZ2 domain of p300, stimulating p300 HAT activity and increasing acetylation levels of recombinant histone H3 and H4 (Reynoird et al., 2010; Shiota et al., 2018). However, the molecular mechanism of how the F1c domain of NUT mediates p300 activation has remained unclear before this study. In this study, we found that the F1c domain consists of TAD1 and TAD2, which engage in bipartite interaction with p300 TAZ2 domain, resulting in p300 auto-acetylation, HAT activation and histone H3K18 and H3K27 hyper-acetylation in chromatin.

C. Q3. “In Fig. 5, the authors stated that BRD4-NUT fusion protein up-regulates ALX1 and promotes EMT, however, the data supporting this claim is thin (pPCR of key genes like SNAIL); there is no data on loss of cell polarity, cell-cell adhesion, gain of migratory and invasive properties, etc. to show that changes in such EMT-related gene expression translates into factual EMT.”

Response: In this study, we focused our investigation on the molecular and structural basis of bipartite binding of BRD4-NUT-TADs and p300-TAZ2 that induces p300 *trans*-activation and histone H3 hyper-acetylation, leading to nuclear condensation on chromatin. We used *ALX1* as an example to illustrate the function of BRD4-NUT/p300 in gene transcription in a model system. To further investigate the transcriptional control of *ALX1* by BRD4-NUT/p300 in NC cells, in the revision, we performed siRNA knockdown of *ALX1* and *BRD4-NUT* in NC 14169 cells, and found that siNUT reduced the transcription of *ALX1*, and both siNUT and siALX1 resulted in increased expression of Involucrin, a marker for differentiation (**Fig. 7d, e**). These new results confirmed the transcriptional regulation of *ALX1* by BRD4-NUT/p300 in NCs. While *ALX1*'s role in EMT has been described in cancers including melanoma, lung and ovarian cancer (Jiao et al., 2019; Yao et al., 2015; Yuan et al., 2013), characterization of its role in EMT in NCs would require a major effort and is beyond the scope of this study. In light of Reviewer #2's constructive comment and our new results, we have modified our discussion in the manuscript.

C. Q4. “The functional consequences of BRD4-NUT/P300 interaction and the resulting hyperacetylation need to be further investigated; additional data points (e.g. resistance to targeted chemotherapy [MS645, JQ1], anti- apoptotic potential, invasion potential, cell cycling [CDK9], cancer stem-cell formation potential) and further discussed is required.”

Response: The functional consequences of BRD4-NUT/p300 interaction and hyper-acetylation have been well characterized by Christ French and colleagues (Eagen and French, 2021; Shiota et al., 2021).

Specifically, it has been shown that BRD4-NUT fusion oncoprotein recruits and activates HAT p300, leading to the formation of hyper-acetylated nuclear foci on chromatin, termed “megadomain” (MD) that is enriched with BRD4-NUT, p300 and hyper-acetylated histones. MDs function as massive super-enhancers, activating transcription of stem cell-related, anti-differentiation transcription factors including *MYC*, *SOX2* and *TP63*. Additionally, chemical inhibitors targeting HDACs and BRD4 bromodomains have been shown to disrupt the megadomain structure. For instance, HDAC inhibitor panobinostat can repress growth and induce differentiation of NC cells through repressing transcriptional expression of *MYC*, suggesting its therapeutic potential for treating NC patients. Indeed, a clinical study reported that two NC patients showed responses to the treatment of FDA-approved HDACi, Vorinostat (SAHA). However, their tumor growth soon recurred, and treatments were stopped due to adverse effects (Maher et al., 2015; Schwartz et al., 2011). *Notably*, our new data revealed that HDACis (SAHA, TMP269 or LMK235) can disperse BRD4-NUT nuclear puncta in LO2 cells overexpressing EGFP-BRD4-NUT-WT shortly after the drug treatment (12-hrs), but the BRD4-NUT puncta formation recurred over time (48 hrs) (**ED-Fig. 5c**). In contrast, BET inhibitor JQ1 and especially MS645 were able to *persistently* disrupt BRD4-NUT puncta formation 4, 12 or 48 hours after the drug treatment (**ED-Fig. 5a,b**). These results indicated that HDACis only transiently affect BRD4-NUT/p300 condensation, whereas BRD4 BrD inhibitors can block BRD4-NUT/p300 condensation in a sustained manner. Further study is warranted to elucidate the molecular dynamics of BRD4-NUT/p300 condensation in chromatin and to explore new ways such as BRD4 inhibition by which its formation can be disrupted. We have enhanced our discussion on this important subject in the revised manuscript.

C. Q5. “*Is the consequent hyperacetylation restricted to H3K27 as shown in Fig. 4c; could other residues (e.g. H3K18) be targeted?*”

Response: To address Reviewer #2’s comment, we have examined H3K18ac in both cellular and *in-vitro* nuclear extract acetylation assays. We found that transfection of BRD4-NUT-WT led to liquid-like co-condensations with H3K18ac and H3K27ac in cell nuclei (**Fig. 4c**), increased H3K18ac and H3K27ac levels in both acetylation assays, and elevated H3K56ac level was only seen in the *in-vitro* nuclear extract assay (**Fig. 5a, b**). The deletion of TAD domains in BRD4-NUT nearly abolished the nuclear co-condensation in cells (**Fig. 4c**) and p300 activation and histone acetylation in both assays (**Fig. 5a,b**). These results confirmed that p300 specifically targets H3K18ac and H3K27ac in chromatin in cells (Shiota et al., 2018), but can non-selectively acetylate multiple lysine residues of recombinant histones *in vitro* (Reynoird et al., 2010).

C. Q6. “*In the diagrammatic representation (Fig. 6d), it is not clear why the authors focused on AXL1 and other anti-differentiation genes despite the fact that data in Fig. 6 shows that BRD4-NUT/P300 interaction alters the expression of other gene sets such as the EMT-related genes.*”

Response: We intended to present a mechanistic model that summarizes the mechanistic details from our study on how BRD4-NUT/p300 work together in aberrant gene transcription in NCs, which has been reported to direct overexpression of stem cell-related and anti-differentiation oncogenic transcription factors such as TP63, SOX2 and MYC. The EMT-related genes are likely downstream effects by the latter TFs, as reported previously (Thompson-Wicking et al., 2013; Yao et al., 2015). In light of Reviewer #2’s helpful comment, we have improved the presentation of this mechanistic model in the manuscript.

D. Reviewer #3 (Remarks to the Author):

It is the first time that the structure of the p300 BRD4-NUT/p300 complex is resolved, providing a novel drug target to block BRD4-NUT/p300 interaction to treat this devastating disease. Besides this, this manuscript mainly confirms the Reynoird et al.’s findings in 2010. In Reynoird’s paper, the authors identified p300 as a binding partner of BRD4-NUT through interaction between p300 TAZ2 domain and BRD4-NUT F1c region. This interaction can activate p300, leading to enhanced HAT activities of p300. Although this manuscript further explored one of the downstream targets ALX1 in the tumorigenesis of NUT carcinoma (NC), most cellular experiments were done in Non-NC models.

Concerning the manuscript itself, I have the following issues that the authors could address.

D. Q1. “I think the oncogenesis of NC relies on the whole aberrant gene transcription landscape, and ALX1 is within this landscape. Although the authors showed EMT induction by wt BRD4-NUT but not by BRD4-NUT with F1c deletion in non-NC models, the direct evidences whether ALX1 is involved are still missing. The authors could simply check if EMT induction by wt BRD4-NUT is compromised by ALX1 depletion. To prove its vital role in NC, the biological functions of ALX1 also needs to be specifically addressed in NC models. For example, ALX1 knockdown experiments could be performed in NC to test its functions in controlling cell proliferation, squamous differentiation and EMT.”

Response: We thank Reviewer #3 for his/her constructive comments and suggestions. In this study, we focused our investigation on the molecular and structural basis of TADs/TAZ2 bipartite binding in BRD4-NUT/p300 that induces p300 *trans*-activation and histone H3 hyper-acetylation, leading to nuclear condensation in chromatin. We used ALX1 as an example to illustrate the BRD4-NUT/p300 function in gene transcription in a model system. To further investigate the transcriptional control of ALX1 by BRD4-NUT/p300 in NC cells, in the revision, we performed siRNA knockdown of ALX1 and BRD4-NUT in NC 14169 cells, and found that siNUT reduced ALX1 transcriptional expression, and both siNUT and siALX1 resulted in increased expression of Involucrin, a marker for differentiation (**Fig. 7d, e**). Our data are consistent with the RNA-seq data showing that ALX1 mRNA levels were decreased by siNUT knockdown in NC PER-403 and TC-797 cells (Morrison-Smith et al., 2020; Schwartz et al., 2011; Thompson-Wicking et al., 2013), and confirm the functional role of ALX1 in NC cell proliferation that is controlled by BRD4-NUT/p300. *Importantly*, our results showed that in the aberrant gene transcription landscape in NC cells, in addition to ALX1, many oncogenic transcription factors including MYC, TP63 and SOX2 that are targets of BRD4-NUT/p300 directly contribute to uncontrolled NC cell proliferation. Considering this reviewer’s helpful comment and our new results, we have modified our discussion in the revised manuscript.

D. Q2. “The overexpression of BRD4-NUT with F1c deletion still forms foci and colocalized with histone hyperacetylated foci (Fig. 4c & d), though to a less extent compared to wt BRD4-NUT. I am wondering if other regions of BRD4-NUT still recruit other HATs. The authors have many BRD4-NUT deletion constructs and it would be interesting to test them in the same setting.”

Response: We appreciate Reviewer #3’s insightful comment. BRD4-NUT TAD1/2 deletion mutants can indeed still form nuclear condensates in chromatin albeit to a lesser extent as compared to BRD4-NUT-WT (**Fig. 4a-c**). This is because as described in our previous study (Han et al., 2020), BRD4 short isoform (BRDS, present in BRD4-NUT) itself can form liquid-liquid phase-separated nuclear condensate in chromatin that is facilitated by its binding to acetylated histones (through the BrDs) as well as to DNA (through the IDRs between the tandem BrDs and between the second BrD and the ET domain). The nuclear condensates formed with BRD4S comprise key transcriptional proteins including BRD4L/S and p300, and mediators and RNA Pol II, capable of facilitating gene transcriptional activation (**ED-Fig. 6**). However, the major distinction between the BRD4S and BRD4-NUT/p300 nuclear condensates is that the latter can prorogate in chromatin via an acetylation-driven mechanism to form super-enhancer-like ‘megadomain’ structure. This can be visualized as discrete nuclear foci in the cell fluorescent imaging analysis. We have enhanced our discussion to reflect this important point in the revision.

D. Q3. “In Figure 4f, the authors observed enhanced H3K27ac, but not H3K56ac and H4K12ac in BRD4-NUT overexpressed cell lysates. But in Figure 4g, the *in vitro* HAT assay showed both enhanced H3K27ac and H3K56ac from the same nuclear extracts. Can the authors explain the discrepancy?”

Response: In the cellular acetylation assay (**Fig. 5a**), we examined p300 HAT activity in chromatin with the transfection of BRD4-NUT (wild type or TAD1/2 deletion mutants) in cells and found that BRD4-NUT-WT increased H3K27ac and H3K18ac (new data) levels, but not H3K56ac and H4K12ac, which is consistent with the known p300 HAT specificity for acetylation of H3K18ac and H3K27ac in chromatin (Reynoird et al., 2010; Shiota et al., 2018). On the other hand, in the *in-vitro* nuclear extract acetylation

assay (**Fig. 5b**), we investigated the activation of p300 HAT activity by BRD4-NUT WT vs. TAD1/2 deletion mutants using recombinant histone H3 with the addition of acetyl-CoA. We found that BRD4-NUT-WT facilitated p300 activation and non-selectively elevated H3K18ac, H3K27ac and H3K56ac levels in recombinant H3 (**Fig. 5b**). Our results indicate that p300 HAT activity specificity is likely influenced by the nucleosome packing in cells. We have improved our discussion to clarify this important point in the manuscript.

E. References:

Candau, R., Scolnick, D.M., Darpino, P., Ying, C.Y., Halazonetis, T.D., and Berger, S.L. (1997). Two tandem and independent sub-activation domains in the amino terminus of p53 require the adaptor complex for activity. *Oncogene* 15, 807-816.

Eagen, K.P., and French, C.A. (2021). Supercharging BRD4 with NUT in carcinoma. *Oncogene* 40, 1396-1408.

Han, X., Yu, D., Gu, R., Jia, Y., Wang, Q., Jaganathan, A., Yang, X., Yu, M., Babault, N., Zhao, C., *et al.* (2020). Roles of the BRD4 short isoform in phase separation and active gene transcription. *Nat Struct Mol Biol* 27, 333-341.

Jiao, J.X., Jiao, L.J., Yang, S., and Zhao, Y.J. (2019). Knockdown of aristaless-like homeobox1 inhibits epithelial-mesenchymal transition through Wnt/beta-catenin signaling pathway in melanoma cells. *Biochem Biophys Res Commun* 511, 105-110.

Krois, A.S., Ferreon, J.C., Martinez-Yamout, M.A., Dyson, H.J., and Wright, P.E. (2016). Recognition of the disordered p53 transactivation domain by the transcriptional adapter zinc finger domains of CREB-binding protein. *Proc Natl Acad Sci U S A* 113, E1853-1862.

Maher, O.M., Christensen, A.M., Yedururi, S., Bell, D., and Tarek, N. (2015). Histone deacetylase inhibitor for NUT midline carcinoma. *Pediatr Blood Cancer* 62, 715-717.

Morrison-Smith, C.D., Knox, T.M., Filic, I., Soroko, K.M., Eschle, B.K., Wilkens, M.K., Gokhale, P.C., Giles, F., Griffin, A., Brown, B., *et al.* (2020). Combined Targeting of the BRD4-NUT-p300 Axis in NUT Midline Carcinoma by Dual Selective Bromodomain Inhibitor, NEO2734. *Mol Cancer Ther* 19, 1406-1414.

Reynoird, N., Schwartz, B.E., Delvecchio, M., Sadoul, K., Meyers, D., Mukherjee, C., Caron, C., Kimura, H., Rousseaux, S., Cole, P.A., *et al.* (2010). Oncogenesis by sequestration of CBP/p300 in transcriptionally inactive hyperacetylated chromatin domains. *EMBO J* 29, 2943-2952.

Schwartz, B.E., Hofer, M.D., Lemieux, M.E., Bauer, D.E., Cameron, M.J., West, N.H., Agoston, E.S., Reynoird, N., Khochbin, S., Ince, T.A., *et al.* (2011). Differentiation of NUT Midline Carcinoma by Epigenomic Reprogramming. *Cancer Research* 71, 2686-2696.

Shiota, H., Alekseyenko, A.A., Wang, Z.A., Filic, I., Knox, T.M., Luong, N.M., Huang, Y., Scott, D.A., Jones, K.L., Gokhale, P.C., *et al.* (2021). Chemical Screen Identifies Diverse and Novel Histone Deacetylase Inhibitors as Repressors of NUT Function: Implications for NUT Carcinoma Pathogenesis and Treatment. *Mol Cancer Res* 19, 1818-1830.

Shiota, H., Barral, S., Buchou, T., Tan, M., Coute, Y., Charbonnier, G., Reynoird, N., Boussouar, F., Gerard, M., Zhu, M., *et al.* (2018). Nut Directs p300-Dependent, Genome-Wide H4 Hyperacetylation in Male Germ Cells. *Cell reports* 24, 3477-3487 e3476.

Thompson-Wicking, K., Francis, R.W., Stirnweiss, A., Ferrari, E., Welch, M.D., Baker, E., Murch, A.R., Gout, A.M., Carter, K.W., Charles, A.K., *et al.* (2013). Novel BRD4-NUT fusion isoforms increase the pathogenic complexity in NUT midline carcinoma. *Oncogene* 32, 4664-4674.

Yao, W., Liu, Y., Zhang, Z., Li, G., Xu, X., Zou, K., Xu, Y., and Zou, L. (2015). ALX1 promotes migration and invasion of lung cancer cells through increasing snail expression. *International journal of clinical and experimental pathology* 8, 12129-12139.

Yuan, H., Kajiyama, H., Ito, S., Yoshikawa, N., Hyodo, T., Asano, E., Hasegawa, H., Maeda, M., Shibata, K., Hamaguchi, M., *et al.* (2013). ALX1 induces snail expression to promote epithelial-to-mesenchymal transition and invasion of ovarian cancer cells. *Cancer Res* 73, 1581-1590.

REVIEWERS' COMMENTS

Reviewer #1 (Remarks to the Author):

The authors have adequately addressed my previous comments.

Reviewer #2 (Remarks to the Author):

The authors have sufficiently addressed all my concerns

Reviewer #3 (Remarks to the Author):

I would like to thank the authors for their adequate and sufficient response to my and all reviewer questions. The manuscript has been clearly improved and I recommend publication.

One minor comment for the discussion section:

"BET BrD and p300/CBP HAT inhibitors have been evaluated in human clinical trials and shown initial efficacy for NC treatment 53-56"

Here, the authors actually do not cite trials inhibition p300 but BET proteins.

To my knowledge p300 inhibition in NC has so far only been shown in preclinical trials: PMID 32366905 and 32371576.

A clinical trial targeting p300/CBP is yet to be starting recruitment: NCT05488548.

Final Revisions (NCOMMS-22-25872A)

1. Reviewer #3

"One minor comment for the discussion section:

"BET BrD and p300/CBP HAT inhibitors have been evaluated in human clinical trials and shown initial efficacy for NC treatment 53-56"

Here, the authors actually do not cite trials inhibition p300 but BET proteins.

To my knowledge p300 inhibition in NC has so far only been shown in preclinical trials: PMID 32366905 and 32371576.

A clinical trial targeting p300/CBP is yet to be starting recruitment: NCT05488548."

Response: We thank the reviewer for his/her careful review of our revised manuscript. The reviewer is correct that only BET BrD inhibitors have been evaluated in human clinical trials for NUT middle carcinoma, but not p300/CBP HAT inhibitors. Note that the new clinical trial NCT05488548 is for a dual BET and p300/CBP BrD inhibitor. Accordingly, we have corrected this sentence in the discussion to: *"BET BrD inhibitors have been evaluated in human clinical studies and shown initial efficacy for NC treatment⁵³⁻⁵⁶."*